# Tensor-Based Joint Beamforming with Ultrasonic and RIS-Assisted Dual-Hop Hybrid FSO mmWave Massive MIMO of V2X

Xiaoping Zhou *, Zhaonan Zeng, Jiehui Li *, Zhen Ma and Le Tong

The College of Information, Mechanical and Electrical Engineering, Shanghai Normal University, Shanghai 200234, China; zengzhaonan2022@163.com (Z.Z.); shnumm@163.com (Z.M.); tongle@shnu.edu.cn (L.T.)
* Correspondence: zxpshnu@163.com (X.Z.); lijh@shnu.edu.cn (J.L.)

**Abstract:** Reconfigurable intelligent surface (RIS)-assisted millimeter-wave (mmWave) communication systems relying on hybrid beamforming structures are capable of achieving high spectral efficiency at a low hardware complexity and with low power consumption. Tensor-based joint beamforming with low-cost ultrasonic and RIS-assisted Dual-Hop Hybrid free space optical (FSO) mm Wave massive Multiple Input Multiple Output (MIMO) of vehicle-to-everything (V2X) is proposed. To address the occlusion problem for high-speed mobility of the vehicle, an RIS-assisted mixed FSO-MIMO V2X system is proposed. The low-cost ultrasonic array signal model is developed to solve the accurate direction-of-arrival (DOA) estimation. The ultrasonic-assisted RIS phase shift matrix based on subspace self-organizing iterations is designed to track the beam direction between RIS and vehicle. Specifically, the associated bandwidth-efficiency maximization problem is transformed into a series of subproblems, where the subarray of phase shifters and RIS elements is jointly optimized to maximize each subarray's rate. The vehicle motion state is transformed into a two-dimensional model for prior distribution to calculate the particle weights of the RIS phase. Multi-vehicle Tucker tensor decomposition is used to describe the high-dimensional beam space. We conceive a multi-vehicle joint optimization method for designing the hybrid beamforming matrix of the base station (BS) and the passive beamforming matrix of the RIS. A cascaded channel decomposition method based on Singular Value Decomposition (SVD) is used to obtain the combined matrix beamforming of BS and vehicle. Our simulation results demonstrate the superiority of the proposed method compared to its traditional counterparts.

**Keywords:** V2X; FSO; joint beamforming; tensor decomposition; massive MIMO





## 1. Introduction

With the increasing demand for faster data rates along with the integration of a wide range of devices into the network, it is necessary to explore new paradigms of wireless communication. Homogeneous physical layer communication, such as radio frequency (RF), laser and optical fiber, are morphing into a hybrid combination to improve the service provided to a large variety of devices [1–3]. A novel downlink satellite communication (SatCom) model was introduced to improve the system performance, in which free space optical (FSO) communication was adopted between a satellite and a high-altitude platform station (HAPS) node [4]. A unified system performance analysis of unmanned aerial vehicle (UAV)-assisted dual-hop FSO/FSO systems with the amplify-and-forward relaying protocol and intensity modulation/direct detection technique was presented [5]. Dual-hop hybrid FSO-RF relay systems combine the advantages of FSO and RF technologies to provide a superior performance. Multiple Input Multiple Output (MIMO) is widely used in RF wireless links in order to alleviate the effect of multi-path fading [6,7]. Similarly, in order to combat the impact of atmospheric turbulence (AT) and pointing errors (PE) on

the FSO link performance, MIMO is also employed in FSO systems [8]. A hybrid FSO/RF system is considered as a Vehicle to Everything (V2X).

V2X plays an important role in vehicle connectivity and autonomous driving. However, due to the high frequency and short wavelength of millimeter wave (mmWave) signals, they are easily blocked by obstacles [9,10]. Therefore, directional antennas are required for beam direction alignment to ensure stable communication quality. To mitigate the effect of path loss, MIMO can generate a high-gain directional beam using beamforming [11,12]. However, obstacles such as buildings and roadside signs can block the line-of-sight (LoS) communication between vehicles and the base station (BS) in mmWave massive MIMO V2X. Therefore, joint beamforming of reconfigurable intelligent surface (RIS)-assisted mmWave massive MIMO V2X can effectively solve problems such as the presence of strong obstructions in the pathway [13–15]. Based on the above studies, a RIS-assisted V2X scheme was proposed to improve the communication range of urban V2X [16]. Since RIS can improve the quality of service (QoS) of wireless communication, a QoS-driven spectrum sharing scheme for RIS-assisted V2X was proposed [17]. The use of real-time software-controlled RIS was investigated to improve the reliability of V2X communication, which can effectively eliminate the multipath fading effect [18]. Considering the congestion under real road conditions, a visible light and RIS-assisted system was proposed to improve the coverage and communication performance of V2X [19]. Moreover, accurate direction-of-arrival (DOA) estimation is crucial in RIS-assisted communication systems.

The one-bit RIS as a signal reflector was introduced to enhance signal transmission in non-line-of-sight (NLOS) situations and substantially simplify the physical hardware for DOA estimation. A non-iterative two-stage channel estimation framework was proposed for DOA estimation in point-to-point RIS-assisted mmWave MIMO systems [20]. To track DOA uncertainty, Bayesian beamforming for a mobile mmWave channel was proposed [21]. The beam tracking problem can be solved by incorporating the Bayesian approach with an expectation maximization (EM) algorithm. An improved genetic algorithm for RIS-assisted beamforming scheme was proposed [22]. The genetic algorithm greatly simplifies the physical hardware for DOA estimation. An atomic parametric-based DOA estimation method for RIS-assisted systems was proposed [23]. System architecture based on RIS-assisted autonomous vehicle localization was investigated, which enabled super-resolution DOA estimation for autonomous vehicles [24]. In addition, there are other studies on DOA estimation for RIS-assisted systems [25–27]. Unlike traditional DOA estimation systems, when tracking the phase of the reflected signal, a low-cost system with a complete DOA receiver is required. Therefore, the DOA estimation problem in V2X needs to be solved urgently. Also, it is difficult to handle the high-dimensional beam space in mmWave massive MIMO V2X.

To decompose high-dimensional beam space, the tensor modeling-based method was proposed for the channel estimation and receiver design of two-hop MIMO V2X communication systems. Accurate high-dimensional channel estimation in highly mobile scenarios became necessary [28]. However, the rank solution corresponding to the canonical polyadic (CP) decomposition is a non-deterministic polynomial (NP) puzzle. A sparse Bayesian tensor of the channel estimation method based on DOA tracking for V2X mmWave massive MIMO systems was proposed [29]. An ultrasonic-assisted tensor channel estimation method for V2X mmWave massive MIMO systems was proposed to improve the safety of cooperative autonomous driving [30]. Ultrasonic-assisted tensor beamforming is a low-cost system capable of decomposing high-dimensional beam space.

In order to solve the above problems, a Tucker tensor-based joint beamforming scheme with ultrasonic and RIS-assisted for dual-hop hybrid FSO-mmWave massive MIMO of V2X is proposed. Firstly, to address the occlusion problem in V2X communication, the RIS-assisted mixed FSO-MIMO V2X system is proposed. The FSO system model and mmWave massive MIMO signal model framework are proposed. Secondly, in order to simultaneously consider solving precise direction of arrival (DOA) estimation and low-cost hardware problems, an ultrasonic array signal model is researched. The beamforming

of RIS-assisted mmWave massive MIMO V2X with low-cost ultrasonics is designed. The DOA information is obtained via accurate measurement of the target's position and angle information. Thirdly, an ultrasonic-assisted RIS phase shift matrix based on subspace self-organizing iterations is designed. Specifically, the associated bandwidth-efficiency maximization problem is transformed into a series of subproblems, where the subarray of phase shifters and RIS elements is jointly optimized to maximize each subarray's rate. The vehicle motion state is transformed into a two-dimensional model for prior distribution to calculate the particle weights of the RIS phase shift matrix. Finally, since the traditional tensor decomposition corresponding to the rank solution is an NP problem, we conceive a multi-vehicle joint optimization method used to design the hybrid beamforming matrix of the BS and the passive beamforming matrix of the RIS. Multi-vehicle Tucker tensor decomposition is used to describe the high-dimensional beam space. A cascaded channel decomposition method based on SVD (Singular Value Decomposition) is used to achieve the combined matrix beamforming of the BS and vehicles. Our simulation results demonstrate the superiority of the proposed method compared to its traditional counterparts.

The rest of the paper is organized as follows. Section 2 introduces the system model. Section 3 presents the ultrasonic and RIS-assisted mmWave massive MIMO V2X joint beamforming problem. Section 4 designs the ultrasonic-assisted RIS phase shift matrix based on subspace self-organizing iterations. Section 5 designs the BS beamforming matrix. In Section 6, the tensor-based design of the user vehicle combination matrix is presented. Section 7 presents the simulation results and, finally, our discussion and summary of the article.

## 2. System Model and Channel Model

In order to combat the impact of AT and PE on the FSO link performance, a block diagram of a mixed FSO-MIMO system is presented in Figure 1, in which the source node (S) communicates with the destination node (D) through a decode-and-forward (DF) relay node (R). In Figure 1, an ultrasonic and RIS-assisted for hybrid FSO-mmWave massive MIMO of V2X system on a highway is proposed. The vehicle speed has an effect on the performance of the system. The increase in speed leads to a larger error, because the increase in vehicle speed produces Doppler shift that affects the beamforming effect. Considering that the LoS communication between the vehicles and the BS is blocked by obstacles and the high-speed mobility of the vehicle, we introduce the one-bit RIS as a signal reflector to enhance signal transmission in NLOS situations and substantially simplify the physical hardware for DOA estimation. The use of RIS-assisted V2X communication with reflected signals can re-establish the communication between the vehicles and the BS. S-R link is equipped with a single antenna, and the R-D link is a mmWave massive MIMO V2X system model. The node R has both optical and RF signal processing capabilities. We employ non-coherent intensity modulation with a direct detection (IM/DD) receiver at R. After converting the incoming optical signal to an electrical signal, node R utilizes a power splitter to separate the alternating current (AC) and direct current (DC) components. The unsolicited DC component (which is normally filtered out at the receiver) is applied to the energy harvesting unit which supplies the harvested power to the RF transmitter. The information-bearing AC component is given to the decoder circuit, which decodes the information, and demodulated using an RF modulation scheme before it is forwarded through the RF transmitter. Figure 2 illustrates the workflow of DF.

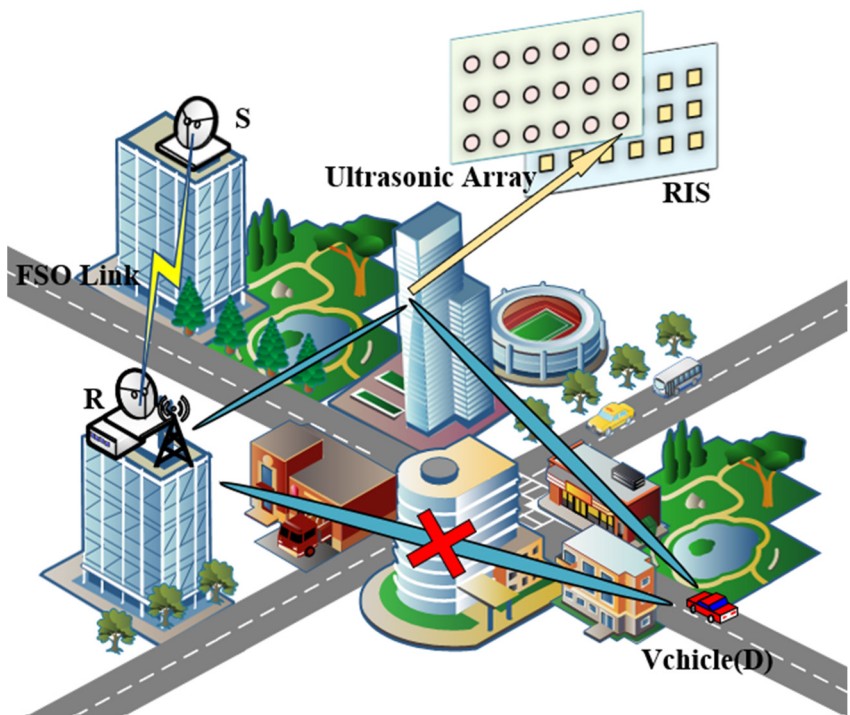

**Figure 1.** Block diagram of ultrasonic and RIS-assisted for Dual-Hop Hybrid FSO-mmWave massive MIMO of V2X system.

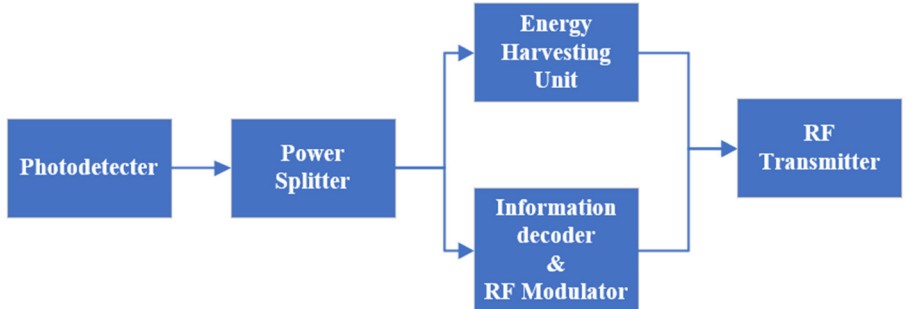

**Figure 2.** DF workflow diagram.

*2.1. FSO System Model*

Since the atmospheric turbulence is generally a medium to strong turbulence case, the Gamma–Gamma model is more consistent with the actual test data in the turbulence case. The FSO channel between S and R is modeled as Gamma–Gamma distribution with pointing error. The statistical behavior of the received optical irradiance is characterized by means of the Gamma–Gamma turbulence model, which has been widely used to model the FSO channel in the recent literature depending on its doubly stochastic scintillation model [31]. The probability distribution function (PDF) of channel coefficient $h_{FSO}$ is given as [32].

$$f_{h_{FSO}}(h_{FSO}) = \frac{\varsigma^2}{h_{FSO}\Gamma(\alpha)\Gamma(\beta)} G_{1,3}^{3,0}\left(\alpha\beta\frac{h_{FSO}}{I_{FSO}M_o}\Big|_{\varsigma^2-1,\alpha-1,\beta-1}^{\varsigma^2}\right) \tag{1}$$

where $\Gamma(.)$ is the well-known Gamma function [33] and $G_{p,q}^{m,n}(x|_{b_1,...,b_q}^{a_1,...,a_p})$ is the Meijer's G-function defined in [34]. The atmospheric turbulence induced fading channel gains of FSO links, denoted as $I_{FSO}$ is modeled with pointing errors and atmospheric attenuation, where $\varsigma = \varpi_e/2\sigma_s$ is the ratio of equivalent beam radius and zero boresight pointing error displacement standard deviation at the photo-detector (PD). The constant term $M_o$ is the power fraction that the detector receives when there is no pointing error [35], and $1/\alpha$ and

$1/\beta$ are the variances of the large and small turbulence eddies, respectively. Parameters $\alpha$ and $\beta$ are the distance-dependent fading variables which correspond to atmospheric turbulence conditions, as described in [35].

$$\alpha = \left\{ \exp\left[ \frac{0.49\sigma_R^2}{\left(1 + 1.11\sigma_R^{\frac{12}{5}}\right)^{\frac{7}{6}}} \right] - 1 \right\}^{-1} \tag{2}$$

$$\beta = \left\{ \exp\left[ \frac{0.51\sigma_R^2}{\left(1 + 0.69\sigma_R^{\frac{12}{5}}\right)^{\frac{5}{6}}} \right] - 1 \right\}^{-1} \tag{3}$$

where $\sigma_R^2 = 1.23C_n^2 v^{7/6} d_{SR}^{11/6}$ denotes the Rytov variance; $C_n^2$, $v$, and $d_{SR}$ represent the refractive index structure constant, wave number, and the link length, respectively. $C_n^2$ usually takes values in the range $10^{-17} - 10^{-13}$ from weak to strong turbulence conditions.

In order to evaluate the performance of the FSO link, we need to derive formulas for the BER and outage probability of the FSO link. For the FSO link with IM/DD detection, the instantaneous SNR is given as [36]

$$\gamma_F = \frac{2P_A^2 R^2}{\sigma_n^2} h_{FSO}^2 \tag{4}$$

where $P_A$ and $R$ represent the average transmitted power and the detector responsivity, respectively; $\sigma_n^2$ is the variance of additive white Gaussian noise with zero mean. The average SNR is defined in [36] as

$$\overline{\gamma_F} = \frac{2P_A^2 R^2}{\sigma_n^2} E\left[h_{FSO}^2\right] = \frac{2P_A^2 R^2}{\sigma_n^2}(\varsigma M_o I_{FSO})^2 \tag{5}$$

According to [36], (1) can be rewritten as a function of $\gamma_F$ instead of $h_{FSO}$, as

$$f_{\gamma_F}(\gamma_F) = \frac{\varsigma^2}{\gamma_F \Gamma(\alpha)\Gamma(\beta)} G_{1,3}^{3,0}\left(\alpha\beta\varsigma\sqrt{\frac{\gamma_F}{\overline{\gamma_F}}}\Big|_{\varsigma^2,\alpha,\beta}^{\varsigma^2+1}\right) \tag{6}$$

### 2.2. Outage Probability of the FSO Link

The outage probability is an important factor in wireless communication. It is defined as the probability that the SNR falls below a certain threshold of quality of service and the FSO link is unable to transmit data. Thus, the FSO outage probability can be expressed as

$$P_{op}(\gamma_{TH}) = P(\gamma_F < \gamma_{TH}) = F(\gamma_{TH}) \tag{7}$$

where $F(\cdot)$ is the cumulative distribution function (CDF) given by

$$F(\gamma_{TH}) = \int_0^{\gamma_{TH}} f_{\gamma_F}(\gamma_F) d\gamma_F \tag{8}$$

where $\gamma_{TH}$ denotes the threshold of the FSO link.

Therefore, the outage probability of the FSO link is derived by substituting (6) into (8). In order to simplify (8), using the integral formula of the Meijer g function [34], the CDF of $\gamma_F$ is given as follows.

$$P_{op}(\gamma_{TH}) = \frac{2^{\alpha+\beta-3}\varsigma^2}{\pi\Gamma(\alpha)\Gamma(\beta)} \times G_{3,7}^{6,1}\left[\left(\frac{\alpha\beta\varsigma}{4}\right)^2 \frac{\gamma_{TH}}{\overline{\gamma_F}} \left| \begin{array}{c} 1, \frac{\varsigma^2+1}{2}, \frac{\varsigma^2+2}{2} \\ \frac{\varsigma^2}{2}, \frac{\varsigma^2+1}{2}, \frac{\alpha}{2}, \frac{\alpha+1}{2}, \frac{\beta}{2}, \frac{\beta+1}{2}, 0 \end{array} \right.\right] \tag{9}$$

### 2.3. BER for the FSO Link

Bit error rate (BER) is an important indicator to evaluate the performance of the FSO link. For an FSO link with IM/DD detection, the average BER is calculated by averaging the conditional BER over the PDF of $f_{h_{FSO}}(h_{FSO})$ defined by

$$P_b = \int_0^\infty P_e f_{h_{FSO}}(h_{FSO}) dh_{FSO} \tag{10}$$

where $P_e$ is the conditional probability given by [37] as

$$P_e = \frac{1}{2} erfc\left(\sqrt{\frac{h_{FSO}^2 \overline{\gamma_F}}{(2M_o I_{FSO}\varsigma)^2}}\right) \tag{11}$$

To simplify the calculation of the integral in (10), the erfc function can be expressed as a Meijer g function [14].

$$erfc(\sqrt{y}) = \frac{1}{\sqrt{\pi}} G_{1,2}^{2,0}\left[y \left| \begin{matrix} 1 \\ 0, 1/2 \end{matrix} \right.\right] \tag{12}$$

Thus, the BER of the FSO link can be expressed as

$$\begin{aligned} P_b(\overline{\gamma_F}) = &\frac{\varsigma^2}{2\sqrt{\pi} h_{FSO} \Gamma(\alpha)\Gamma(\beta)} \times \int_0^\infty G_{1,2}^{2,0}\left[\frac{\overline{\gamma_F}}{(2M_o I_{FSO}\varsigma)^2} h_{FSO}^2 \left| \begin{matrix} 1 \\ 0, 1/2 \end{matrix} \right.\right] \\ &\times G_{1,3}^{3,0}\left[\alpha\beta \frac{h_{FSO}}{I_{FSO} M_o} \left| \begin{matrix} \varsigma^2 \\ \varsigma^2 - 1, \alpha - 1, \beta - 1 \end{matrix} \right.\right] dh_{FSO} \end{aligned} \tag{13}$$

To simplify the BER of the FSO link, using the integral formula of the product of two Meijer g functions [34], the BER of the FSO link can be rewritten as

$$P_b(\overline{\gamma_F}) = \frac{2^{\alpha+\beta-4}\varsigma^2}{\pi^{3/2}\Gamma(\alpha)\Gamma(\beta)} \times G_{7,4}^{2,6}\left[\frac{4\overline{\gamma_F}}{(\alpha\beta\varsigma)^2} \left| \begin{matrix} \frac{1-\varsigma^2}{2}, \frac{2-\varsigma^2}{2}, \frac{1-\alpha}{2}, \frac{2-\alpha}{2}, \frac{1-\beta}{2}, \frac{2-\beta}{2} \\ 0, \frac{1}{2}, \frac{-\varsigma^2}{2}, \frac{1-\varsigma^2}{2} \end{matrix} \right.\right] \tag{14}$$

At high SNR and similar to the CDF, the BER of the FSO link for high SNR can be given asymptotically as described in [13].

$$\begin{aligned} P_b(\overline{\gamma_F} \to \infty) = &\frac{2^{\alpha+\beta-4}\varsigma^2}{\pi^{3/2}\Gamma(\alpha)\Gamma(\beta)} \sum_{k=1}^6 z^{a_k-1} \\ &\times \frac{\Pi_{l=1,l\neq k}^6 \Gamma(a_k-a_l)\Gamma(1+b_1-a_k)\Gamma(1+b_2-a_k)}{\Pi_{l=3}^4 \Gamma(a_k-b_l)\Gamma(2-a_k)} \end{aligned} \tag{15}$$

Here, some of these parameters are expressed as follows:

$$z = \frac{(\alpha\beta\varsigma)^2}{4\overline{\gamma_F}} \tag{16}$$

$$a_k = \left\{\frac{1-\varsigma^2}{2}, \frac{2-\varsigma^2}{2}, \frac{1-\alpha}{2}, \frac{2-\alpha}{2}, \frac{1-\beta}{2}, \frac{2-\beta}{2}\right\}, for\ k \in \{1, \ldots, 6\} \tag{17}$$

$$b_l = \left\{0, \frac{1}{2}, \frac{-\varsigma^2}{2}, \frac{1-\varsigma^2}{2}\right\}, for\ l \in \{1, \ldots, 4\} \tag{18}$$

### 2.4. DF Transmission Protocal

The communication between S and D is accomplished in two time slots, $T_1$ and $T_2$, for the first and second hop, respectively. The relay harvests energy during $T_1$ only, as simultaneous harvesting and discharging of power increases the complexity at the node. The S employs sub-carrier intensity modulation (SIM) to convert RF signal vector *r* with

electrical power $\delta$ to an optical signal. A DC bias $A \in [A_{\min}, A_{\max}]$ (where $A_{\min}$ and $A_{\max}$ are minimum and maximum values of DC bias, respectively) is added to the RF signal to ensure a non-negative optical signal. Let $P_s$ represent the electrical power of S transmitting the optical signal vector $\boldsymbol{s_1}$; then, we can write

$$\boldsymbol{s_1} = \sqrt{P_s}[\delta r + A] \tag{19}$$

where $\delta$ is the electrical to optical conversion coefficient. In order to prevent clipping due to non-linearity of the laser diode, such that it operates in the linear region, $\delta$ should satisfy the following constraint [31].

$$\delta \leq \min\left(\frac{A - A_{\min}}{\rho}, \frac{A_{\max} - A}{\rho}\right) \tag{20}$$

The electrical signal at the output of the PD can be expressed as

$$y_{FSO} = h_{FSO}\boldsymbol{s_1} + n_{FSO} \tag{21}$$

where $h_{FSO} = (P_s^2 \eta^2 / \sigma_{FSO}^2) I_{FSO}$ is the channel coefficient of S-D link, where $\eta$ is the optical to electrical conversion coefficient. The noise $n_{FSO}$ is due to circuit noise as well as high-intensity background illumination and is conventionally modeled as being zero mean additive white Gaussian noise. $\sigma_{FSO}^2$ is the variance of additive white Gaussian noise with zero mean.

In addition, since BER and outage probability are derived from atmospheric turbulence and pointing errors, we propose a method to estimate the atmospheric channel coefficient of S-D link $h_{FSO}$ using BER and outage probability. The atmospheric channel coefficient estimation model is shown in Figure 3. We decompose the electrical signal at the output of the PD $y_{FSO}$ using light gradient boosting machine (LightGBM) and then input the BER and outage probability together into the bi-directional long short-term memory (BiLSTM) to estimate the atmospheric channel coefficient.

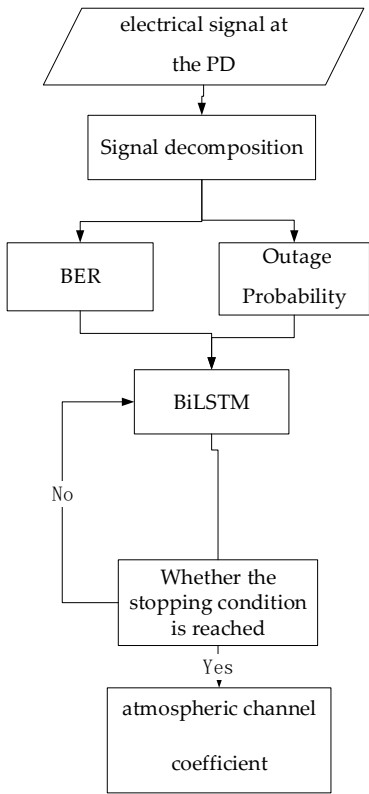

**Figure 3.** Atmospheric channel coefficient estimation model.

BiLSTM is a combination of forward LSTM and backward LSTM, which uses a bidirectional input structure to capture sequence features from both positive and negative directions, allowing it to learn the effective information of features from multiple perspectives. BiLSTM has the same basic structure as LSTM, including an input gate, forgetting gate, output gate and memory unit, as shown in Figure 4.

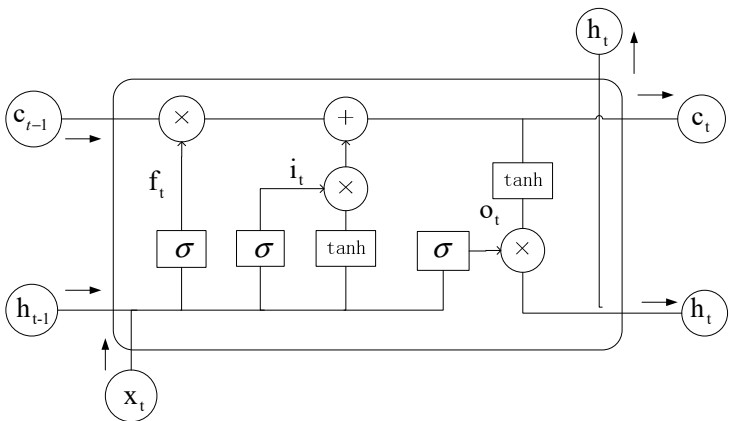

**Figure 4.** LSTM structure diagram; the arrows indicate the direction of data transfer.

The function of the input gate is to control the input of new information, and the computational principle is as follows:

$$i_t = \sigma(w_i[h_{t-1}, x_t] + b_i) \tag{22}$$

where $h_{t-1}$ is the output vector of the LSTM at the previous moment, $x_t$ is the input vector of the LSTM at the current moment, $[h_{t-1}, x_t]$ represents the concatenation of the two vectors, $w_i$, $b_i$ and $i_t$ are the weight vector, bias vector and calculation result of the input gate, respectively, and $\sigma$ is the Sigmoid function.

The function of the forgetting gate is to control the retention and forgetting of information and it is calculated as follows:

$$f_t = \sigma(w_f[h_{t-1}, x_t] + b_f) \tag{23}$$

$$c_t = f_t \odot c_{t-1} + i_t \tanh \odot (w_c[h_{t-1}, x_t] + b_c) \tag{24}$$

where $w_f$, $b_f$ and $f_t$ are the weight vector, bias vector, and calculation result of the forgetting gate, respectively; $c_{t-1}$ is the memory unit of the previous moment, $\odot$ refers to the point-by-point multiplication of the elements in the vector, $w_c$ and $b_c$ are the weight vector and bias vector of the memory unit, respectively, and $c_t$ is the memory unit of the current moment.

The function of the output gate is to control the output of the information in the memory unit, and the calculation principle is as follows:

$$o_t = \sigma(w_o[h_{t-1}, x_t] + b_o) \tag{25}$$

$$h_t = o_t \odot \tanh(c_t) \tag{26}$$

where $w_o$, $b_o$, and $o_t$ are the weight vector, bias vector, and calculation result of the output gate, respectively; $h_t$ is the output vector of the LSTM unit at the current moment. The BiLSTM is an improved structure of LSTM, including forward LSTM and backward LSTM, which can extract bi-directional features of time series data from both forward and backward directions to obtain better results. The structure of BiLSTM is shown in Figure 5.

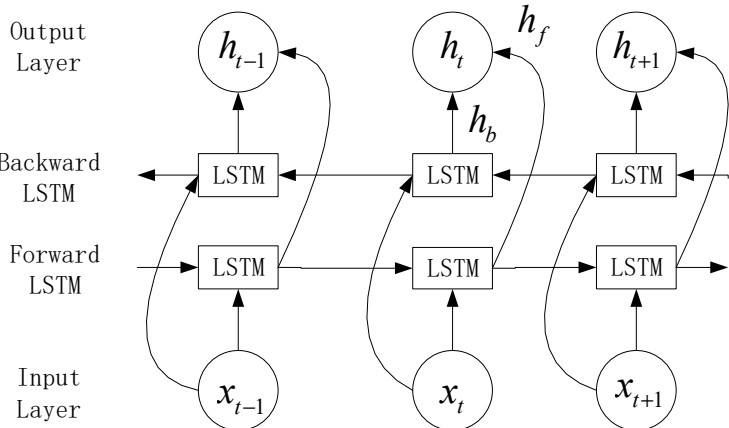

**Figure 5.** BiLSTM structure diagram; the arrows indicate the direction of data transfer.

For the BiLSTM, the forward network output $\overrightarrow{h_t}$ and the backward network output $\overleftarrow{h_t}$ are calculated using (23)–(26), respectively, and then the corresponding elements of these two outputs are concatenated to produce the final output $h_{FSO}$.

$$h_{FSO} = [\overrightarrow{h_t}, \overleftarrow{h_t}] \tag{27}$$

*2.5. RIS-Assisted mmWave Massive MIMO System Model*

As shown in Figure 1, the BS-vehicle link is blocked due to the presence of obstructions. The BS can only communicate through the BS-RIS-vehicle link, which is equipped with hybrid beamforming technology and relies on $N_t$ transmit antennas and $M_t$ RF chains. At the vehicle, a structure with $N_r$ receive antennas and $M_r$ RF chains is used.

The transmit signals of the BS have $N_s$ data streams $s_2 = [s_1, s_2, \ldots, s_{N_s}]^T \in \mathbb{C}^{N_s \times 1}$. At the BS, the transmit signals are firstly precoded by the baseband digital beamformer $F_{BB} \in \mathbb{C}^{M_t \times N_s}$. Then, the signal $F_{BB}s_2$ is precoded by the analogue beamformer $F_{RF} \in \mathbb{C}^{N_t \times M_t}$. Let $s_2 = \sqrt{P_R}y_{FSO}$ denote the signal forwarded by the node R and $F_{all} = F_{RF}F_{BB}$ be the beamforming matrix at BS, where $P_R$ denotes the hybrid mmWave massive MIMO chain transmit power. Then, the transmit signal at the $n$-th time slot after mixing analogue and digital beamforming can be expressed as

$$X(n) = \sqrt{\frac{P_t}{N_s}}F_{all}s_2 \tag{28}$$

where $P_t$ is the transmit power.

Let $H_{all} = H_{RU}\Phi H_{BR}$ be the signals through the BS-RIS-vehicle cascade channel, where $H_{BR} \in \mathbb{C}^{N_{RIS} \times N_t}$ is the BS-RIS channel, $H_{RU} \in \mathbb{C}^{N_r \times N_{RIS}}$ is the RIS-vehicle channel, and $\Phi = diag[e] \in \mathbb{C}^{N_{RIS} \times N_{RIS}}$ is the phase shift matrix of the RIS with $N_{RIS}$ reflecting elements. $e = (e^{j\phi_1}, e^{j\phi_2}, \ldots, e^{j\phi_{N_{RIS}}})$ is the vector consisting of the reflection coefficients of each RIS reflecting element.

The signals passing through the cascade channel reach the vehicle, which applies a combination matrix $W_{all} = W_{RF}W_{BB}$. $W_{RF} \in \mathbb{C}^{N_r \times M_r}$ is an analogue combiner and $W_{BB} \in \mathbb{C}^{M_r \times N_r}$ is a digital combiner. Then, the signals received by the vehicle at D can be expressed as

$$Y(n) = \sqrt{\frac{P_t}{N_s}}W_{all}^H H_{all}F_{all}s_2 + N \tag{29}$$

where $N \sim \mathcal{CN}(0, \sigma^2 I_{N_r}) \in \mathbb{C}^{N_r \times 1}$ denotes additive Gaussian white noise.

According to [38], we can further express the achievable spectral efficiency as

$$R = \log_2\left(\det\left|I_{N_r} + \frac{P_t}{\sigma^2 N_s}W_{all}^H H_{all} F_{all} F_{all}^H H_{all}^H W_{all}\right|\right) \tag{30}$$

*2.6. Ultrasonic Array Signal Model*

As shown in Figure 1, the vehicle communicates through the RIS reflection signals in V2X communication when the direct link is blocked. Then, we consider the ultrasonic array stacked parallel to the RIS plane, and the ultrasonic sensor array elements correspond to the positions of the RIS elements one by one. The ultrasonic sensor spacing is half a wavelength. At the moment $k$, $N_V$ narrowband ultrasonic signals from the vehicle are incident to the $N_{RIS}$ receiving array elements on the ultrasonic array. The two-dimensional incoming wave direction of the $q$-th ultrasonic signal $\hat{x}_q(k)$ is defined as $\boldsymbol{\vartheta}_q(k) = [\beta_q(k), \varphi_q(k)]$. $\beta_q(k) \in (-\pi/2, \pi/2)$ and $\varphi_q(k) \in (0, \pi/2)$ are defined as the azimuth and pitch angles, respectively. Then, the ultrasonic signal received on the ultrasonic array is expressed as

$$\hat{y}'(k) = \sum_{q=1}^{N_V}\left\{[\boldsymbol{b}(\varphi_q(k)) \otimes \overline{\boldsymbol{v}}(\boldsymbol{\vartheta}_q(k))]\hat{x}_q(k)\right\} + \hat{\boldsymbol{n}}'(k) \tag{31}$$

where $\boldsymbol{b}(\varphi_q(k))$ is the guiding vector of the array, $\overline{\boldsymbol{v}}(\boldsymbol{\vartheta}_q(k)) = [1, \boldsymbol{v}_q^T(k)]^T$; $\boldsymbol{v}_q(k)$ is the unit direction vector pointing to the target emission source, as shown in (32). The symbol $\otimes$ denotes the Kronecker product and $\hat{\boldsymbol{n}}'(k)$ is the Gaussian noise generated by the process.

$$\boldsymbol{v}_q(k) = \left[\cos\beta_q(k)\cos\varphi_q(k), \sin\beta_q(k)\cos\varphi_q(k), \sin\varphi_q(k)\right]^T \tag{32}$$

$N_u$ snapshots are taken and the ultrasonic output signal received by the ultrasonic array at moment $k$ is represented as

$$\hat{Y}'(k) = \boldsymbol{\Psi}\hat{X}^T(k) + \hat{N}'(k) \tag{33}$$

where $\boldsymbol{\Psi} = [\boldsymbol{\Psi}(\boldsymbol{\vartheta}_1(k)), \boldsymbol{\Psi}(\boldsymbol{\vartheta}_2(k)), \ldots, \boldsymbol{\Psi}(\boldsymbol{\vartheta}_{N_V}(k))] \in \mathbb{C}^{N_{RIS} \times N_V}$ is the directional matrix of the ultrasonic array, $\boldsymbol{\Psi}(\boldsymbol{\vartheta}_q(k)) = \boldsymbol{b}(\varphi_q(k)) \otimes \overline{\boldsymbol{v}}(\boldsymbol{\vartheta}_q(k))$, $\hat{X}(k) = [\hat{x}_1(k), \hat{x}_2(k), \ldots, \hat{x}_{N_V}(k)] \in \mathbb{C}^{N_u \times N_V}$ is the ultrasonic signal matrix, and $\hat{N}'(k) = \left[\hat{\boldsymbol{n}}_1'(k), \hat{\boldsymbol{n}}_2'(k), \ldots, \hat{n}_{N_u}'(k)\right] \in \mathbb{C}^{N_{RIS} \times N_u}$ is the Gaussian noise matrix.

*2.7. MmWave Channel Model*

According to [28], we define the BS-RIS channel and the RIS-vehicle channel in the V2X communication reflection link as

$$\boldsymbol{H}_{BR} = \sqrt{\frac{N_t N_{RIS}}{L_{BR}}}\sqrt{A_{BR}}\sum_{l=1}^{L_{BR}}\alpha_{BR}^{(l)}\boldsymbol{a}_{RIS}(\phi_{BR,r}^{(l)}, \theta_{BR,r}^{(l)})\boldsymbol{a}_{BS}^H(\phi_{BR,t}^{(l)}, \theta_{BR,t}^{(l)})e^{2\pi f_d n T_s} \tag{34}$$

$$\boldsymbol{H}_{RU} = \sqrt{\frac{N_r N_{RIS}}{L_{RU}}}\sqrt{A_{RU}}\sum_{l=1}^{L_{RU}}\beta_{RU}^{(l)}\boldsymbol{a}_{UE}(\phi_{RU,r}^{(l)}, \theta_{RU,r}^{(l)})\boldsymbol{a}_{RIS}^H(\phi_{RU,t}^{(l)}, \theta_{RU,t}^{(l)})e^{2\pi f_d n T_s} \tag{35}$$

where $L_{BR}$ and $L_{RU}$ are the total number of signal paths in the BS-RIS and RIS-vehicle channel, respectively. $A_{BR}$ and $A_{RU}$ represent the path loss in the BS-RIS and RIS-vehicle channel, respectively. $\alpha_{BR}^{(l)}, \beta_{RU}^{(l)} \sim \mathcal{CN}(0, 1)$ are the composite path gain in the BS-RIS and RIS-vehicle channel, respectively. $\theta_{BR,t}^{(l)}$ and $\phi_{BR,t}^{(l)}$ are the azimuth and elevation angles of the angle of departure (AOD) of the $l$-th path in the BS-RIS channel, $\theta_{BR,r}^{(l)}$ and $\phi_{BR,r}^{(l)}$ are the azimuth and elevation angles of the angle of arrival (AOA) of the $l$-th path in the BS-RIS channel. Similar, $\theta_{RU,t}^{(l)}(\theta_{RU,r}^{(l)})$ and $\phi_{RU,t}^{(l)}(\phi_{RU,r}^{(l)})$ are the azimuth and elevation angles of the AOD (AOA) of the $l$-th path in the RIS-vehicle channel. $f_d$ is the Doppler frequency,



and $T_s$ is the system sampling period. $\boldsymbol{a}_o, o \in \{BS, RIS, UE\}$ is the BS, RIS or vehicle array guidance vector, respectively.

Uniform planar array (UPA) antennas are used at BS, RIS, and vehicle. Therefore, the typical array guidance vector for UPA antennas can be expressed as

$$\boldsymbol{a}_o(\phi, \theta) = \frac{1}{\sqrt{UW}} \left[ 1, \ldots, e^{j\frac{2\pi}{\lambda}d(u\sin\phi\sin\theta + w\cos\theta)}, \ldots, e^{j\frac{2\pi}{\lambda}d((U-1)\sin\phi\sin\theta + (W-1)\cos\theta)} \right]^T \quad (36)$$

where $\lambda$ is the signal wavelength, $d$ is the adjacent antenna spacing, generally taking mmWave half wavelength, and $u \in \{0, 1, \ldots, U-1\}$, $w \in \{0, 1, \ldots, W-1\}$, $U$, and $W$ are the number of antennas in horizontal and vertical directions, respectively.

## 3. Ultrasonic and RIS-Assisted mmWave Joint Beamforming Problem for Massive MIMO V2X

In this section, beamforming design is performed for ultrasonic and RIS-assisted mmWave massive MIMO V2X systems. Considering the maximum transmit power of the BS and the unit-mode constraint of the RIS passive reflection elements, a joint optimization of the phase shift matrix of the RIS $\boldsymbol{\Phi}$, the beamforming matrix of the BS $\boldsymbol{F}_{all}$ and the combined matrix of the vehicle $\boldsymbol{W}_{all}$ are performed to maximize the spectral efficiency.

The joint beamforming problem modeled in this paper is represented as

$$\max_{\boldsymbol{W}_{all}, \boldsymbol{F}_{all}, \boldsymbol{\Phi}} \quad \log_2 \left( \det \left| \boldsymbol{I}_{N_r} + \frac{P_t}{\sigma^2 N_s} \boldsymbol{W}_{all}^H \boldsymbol{H}_{all} \boldsymbol{F}_{all} \boldsymbol{F}_{all}^H \boldsymbol{H}_{all}^H \boldsymbol{W}_{all} \right| \right)$$

$$\text{s. t. } \boldsymbol{H}_{all} = \boldsymbol{H}_{RU} \boldsymbol{\Phi} \boldsymbol{H}_{BR}$$
$$\text{tr}\left( \boldsymbol{F}_{all} \boldsymbol{F}_{all}^H \right) \leqslant P_{\max} \quad\quad (37)$$
$$\boldsymbol{\Phi} = \text{diag}(\boldsymbol{e})$$
$$|\boldsymbol{e}(i)| = 1, \ i = 1, 2, \cdots, N_{RIS}$$

where $P_{\max}$ is the maximum transmit power of BS and $\boldsymbol{e}(i)$ is the $i$-th element in vector $\boldsymbol{e}$.

Since $\boldsymbol{W}_{all}$, $\boldsymbol{F}_{all}$ and $\boldsymbol{\Phi}$ are mutually coupled in the objective function and the joint beamforming problem of the above equation is nonconvex, a direct calculation will be difficult to solve. In order to simplify the above problem, we can solve the problem by alternating optimization, i.e., fixing several other matrices when solving one matrix, followed by alternating updates of the matrices to be solved in several iterations. Therefore, we will design the phase shift matrix of RIS $\boldsymbol{\Phi}$, the beamforming matrix of BS $\boldsymbol{F}_{all}$, and the combination matrix of the vehicle $\boldsymbol{W}_{all}$ separately. For the RIS-assisted MIMO V2X system, a non-convex optimization function is constructed with the objective of maximum spectral efficiency, and there is no optimal algorithm. Therefore, this paper then proposes a beamforming design based on alternating iterations using the characteristic that the qualifying conditions of the optimization function are not coupled with each other. Specifically, the scheme is based on alternating iterations to achieve suboptimal design of the RIS phase shift matrix, BS beamforming matrix, and vehicle combination matrix in order to solve the optimization problem. In the design of the RIS phase shift matrix, this paper combines the subspace self-organizing iterative algorithm to transform the nonconvex optimization problem into a series of easily solvable subproblems. Numerical simulation results show that the scheme in this paper achieves a better compromise in terms of spectral efficiency improvement compared with existing schemes.

## 4. Ultrasonic-Assisted RIS Phase Shift Matrix Design Based on Subspace Self-Organizing Iterations

Since vehicles are constantly moving on the road, next, we will consider the motion state of the vehicle and the transmission of the communication signals, then design the RIS phase shift matrix in this way.

### 4.1. RIS Initial Phase Shift Matrix Solving Based on Ultrasonic-Assisted PF Algorithm

We estimate the ultrasonic DOA arriving at the ultrasonic array and design a rough phase shift matrix of RIS accordingly. The particle filter (PF) algorithm is an algorithm for real-time localization and tracking, and its goal is to recursively update the results by continuously updating the particle weights based on the measurement model and the state transition model. In this algorithm, the observation likelihood function can be expressed as

$$P(\hat{Y}'(k)|\varsigma_i(k)) = \left\{ \frac{\widetilde{P}(\hat{Y}'(k)|\varsigma_i(k))}{\max_{\varsigma_i(k)} \widetilde{P}(\hat{Y}'(k)|\varsigma_i(k))} \right\}^\omega \tag{38}$$

where $\omega$ is the exponential factor determined by the simulation experiment [25]. $\widetilde{P}(\hat{Y}'(k)|\varsigma_i(k))$ denotes the likelihood function determined by the maximum likelihood estimation method. $\varsigma_i(k)$ is the particle, and $i = 1, 2, \ldots, \rho$; the number of particles is $\rho$. The initial weight of each particle is $1/\rho$, and the initial value is generated randomly. Since the conventional likelihood function is greatly affected by environmental conditions such as a low signal-to-noise ratio, the update of particles will become difficult. We apply the propagation operator method in MUSIC spectrum estimation to improve the likelihood function. The covariance matrix of the ultrasonic transducer array output data in (39) is expressed as

$$\Im = E[\hat{Y}'(k)\hat{Y}'H(k)] \tag{39}$$

The covariance matrix is chunked into the following form:

$$\Im = \begin{bmatrix} \Im_1 \\ \Im_2 \end{bmatrix} \tag{40}$$

where $\Im_1$ and $\Im_2$ denote $N_r \times N_r$ and $(N_t - N_r) \times N_r$ dimensional matrices, respectively. $\Im_1$ is a non-singular matrix with linear independent column vectors. According to the propagation operator method, when noise is not considered, there exists a matrix that acts as a propagation operator, such that

$$\Im_2 = \Xi^H \Im_1 \tag{41}$$

Considering the presence of Gaussian noise in the real situation and the finite length of the array output, the problem of solving the propagation operator matrix $\Xi$ can be formulated as the following minimization problem:

$$\min_{\Omega} ||\Im_2 - \Xi^H \Im_1||_F \tag{42}$$
$$\text{s.t. } \Im_2 \to \Xi^H \Im_1$$

The propagation operator matrix $\Xi$ can be found as follows:

$$\Xi = (\Im_1 \Im_1{}^H)^{-1} \Im_1{}^H \Im_2 \tag{43}$$

Construct the matrix $\Gamma = [\Xi^H, I_{N_t-N_r}]$, where $I_{N_t-N_r}$ is a $N_t - N_r$ dimensional unit matrix, such that it satisfies the following condition

$$\Gamma^H \Im = 0 \tag{44}$$

We use the idea of the MUSIC algorithm, i.e., the signal is orthogonal to the noise subspace so that the spectral function can be estimated. As shown in (44), we can see that the matrix is orthogonal to the noise subspace, so we can construct the following spectral function:

$$\Phi(\boldsymbol{\vartheta}_m(k)) = \frac{1}{\boldsymbol{\Psi}^H(k)\Gamma\Gamma^H\boldsymbol{\Psi}(k)} \tag{45}$$

It can be seen that the matrix $\boldsymbol{\Gamma}$ contains the observed information, while the angular information $\boldsymbol{\vartheta}_m(k)$ contains priori information, so the spectral function in (39) can be used to improve the likelihood function and update the particle weights; the likelihood function in (38) can be expressed as

$$\widetilde{P}(\hat{Y}'(k)|\varsigma_i(k)) = \Phi(\boldsymbol{\vartheta}_m(k)) \tag{46}$$

In this algorithm, the prior distribution $P(\varsigma_i(k)|\varsigma_i(k-1))$ is obtained according to the state transition model $\aleph(k)$

$$P(\varsigma_i(k)|\varsigma_i(k-1)) = f\left\{ \sum_{j=1}^{J} \left( \aleph_j(k) \frac{p_j^*(\aleph_j(k))}{\sum\limits_{j=1}^{J} p_j^*(\aleph_j(k))} \right) \right\} \tag{47}$$

where $\aleph_j(k)$ is the $j$-th possible motion state of the vehicle at the moment $k$, $J$ is the total number of motion states, $p_j^*(\aleph_j(k))$ is the probability of this state occurring, and $f(\cdot)$ is the mapping relationship. Equation (47) updated the results $\aleph_j(k)$ in real time by continuously updating the particle weights based on the measurement model and the state transition model.

The particle $\varsigma_i(k)$ is updated by the prior distribution

$$\varsigma_i(k) \sim P(\varsigma_i(k)|\varsigma_i(k-1)) \tag{48}$$

The formula for calculating the particle weights in the update phase is expressed as

$$\begin{aligned} \overline{w}_i(k) &= \overline{w}_i(k-1) \frac{P(\hat{Y}'|\varsigma_i(k))P(\varsigma_i(k)|\varsigma_i(k-1))}{P(\varsigma_i(k)|\varsigma_i(k-1))} \\ &= \overline{w}_i(k-1)P(\hat{Y}'|\varsigma_i(k)) \end{aligned} \tag{49}$$

The normalized weight is expressed as

$$\hat{w}_i(k) = \frac{\overline{w}_i(k)}{\sum\limits_{i=1}^{\rho} \overline{w}_i(k)} \tag{50}$$

After a series of iterations, some particles that tend to degenerate are generated, i.e., the weights of the particles become very small. To solve this problem, it is necessary to resample at the appropriate time. The number of particles should be kept constant and the weight of each particle rescaled to its initial value $1/\rho$.

Let $\psi_q(k)$ be a one-dimensional DOA result, and the angular transformation relation with two-dimensional DOA is

$$\begin{cases} \cos\psi_q(k) = \cos\beta_q(k)\cos\varphi_q(k) \\ \sin^2\psi_q(k) = \sin^2\beta_q(k)\cos^2\varphi_q(k) + \sin^2\varphi_q(k) \end{cases} \tag{51}$$

Thus, the DOA tracking result of the $m$-th ultrasonic array element can be expressed as

$$\psi_m(k) = \sum_{i=1}^{\rho} \hat{w}_i(k)\varsigma_i(k) \tag{52}$$

Since the DOA ultrasonic information reflects the relative position relationship between the vehicle and the joint ultrasonic-RIS array, we take the most representative angle in the tracking results and also consider the azimuth information of BS relative to RIS based on the principle of specular reflection; thus, we can obtain a rough phase shift matrix of RIS $\boldsymbol{\Phi}_0 = diag\left[e^{j\phi_0}, e^{j\phi_0}, \cdots, e^{j\phi_0}\right] \in \mathbb{C}^{N_{RIS} \times N_{RIS}}$.

*4.2. State Transition Model of Vehicle Motion for Prior Distribution*

The observation model $\aleph(k)$ of vehicle motion for prior distribution is calculated in this section. The PF algorithm in Section 4.1 is the algorithm used for real-time localization and tracking. The results are recursively updated by continuously updating the particle weights based on the measurement model and the state transition model. According to Newtonian mechanics of motion, if the vehicle moves in a fixed direction with initial position $S_0$ and initial velocity $v_0$, the acceleration $a$ in a short time interval $\tau$ can be considered as a constant value. Then, the position $S_1$ and velocity $v_1$ after the motion can be expressed as

$$\begin{cases} S_1 = S_0 + v_0\tau + \frac{1}{2}a\tau^2 \\ \quad v_1 = v_0 + a\tau \end{cases} \tag{53}$$

Due to the uncertainty of vehicle motion, uniform or uniformly variable speed motion models cannot accurately describe the actual motion of the vehicle. The current statistics (CS) model takes into account the acceleration variation of the moving target. The acceleration change of the target at the next moment is considered to be within the acceleration neighborhood of the previous moment [39]. The model that takes acceleration into account is clearly more suitable for V2X communication. Let the sampling interval be $\tau$ and the motion state of the target at sampling moment $k\tau$ be represented by vector $[S(k), v(k), a(k)]^T$. Then, the motion state $[S(k+1), v(k+1), a(k+1)]^T$ of the target at the next sampling moment $(k+1)\tau$ can be represented by the CS model as

$$\begin{bmatrix} S(k+1) \\ v(k+1) \\ a(k+1) \end{bmatrix} = \widetilde{\boldsymbol{\lambda}} \begin{bmatrix} S(k) \\ v(k) \\ a(k) \end{bmatrix} + \widetilde{\boldsymbol{\hbar}}\overline{a} + \widetilde{\boldsymbol{N}}(k) \tag{54}$$

where $\widetilde{\boldsymbol{\lambda}}$ is the state transfer matrix of the target motion, $\widetilde{\boldsymbol{\hbar}}$ is the control matrix, $\overline{a}$ is the acceleration average, and $\widetilde{\boldsymbol{N}}(k)$ is the Gaussian noise matrix.

$$\widetilde{\boldsymbol{\lambda}} = \begin{bmatrix} 1 & \tau & (e^{-\mu\tau} + \mu\tau - 1)/\mu^2 \\ & 1 & (1 - e^{-\mu\tau})/\mu \\ & & e^{-\mu\tau} \end{bmatrix} \tag{55}$$

$$\widetilde{\boldsymbol{\hbar}} = \begin{bmatrix} \frac{1}{\mu}(-\tau + \frac{\mu\tau^2}{2} + \frac{1-e^{-\mu\tau}}{\mu}) \\ \tau - \frac{1-e^{-\mu\tau}}{\mu} \\ 1 - e^{-\mu\tau} \end{bmatrix} \tag{56}$$

where $\mu$ is the maneuver frequency.

Since a vehicle does not always move in a straight line in a fixed direction, we extend the vehicle movement to a two-dimensional plane to better fit the actual movement of the vehicle on the road. Using a two-dimensional right-angle coordinate system as a reference, let the $x$-axis be in the direction the vehicle's head is pointing and the $y$-axis be perpendicular to the $x$-axis.

Considering that the speed of the vehicle changing lanes does not change significantly in real situations, the motion of the vehicle on the $y$-axis is approximated as uniform motion for simplicity. Therefore, the vehicle motion state vector at sampling moment $k\tau$ can be reformulated as

$$\aleph(k) = [S_x(k), v_x(k), a_x(k), S_y(k), v_y(k)]^T \tag{57}$$

where $S_x(k)$, $v_x(k)$ and $a_x(k)$ are the position, velocity and acceleration of the vehicle in the $x$-axis direction at the sampling moment $k\tau$, respectively. $S_y(k)$ and $v_y(k)$ are the position and velocity of the vehicle in the $y$-axis direction at sampling moment $k\tau$, respectively.

Then, the vehicle motion state transition model in V2X communication scenario is represented by the following equation

$$\aleph(k+1) = \boldsymbol{\lambda}\aleph(k) + \hbar\bar{a} + \overline{N}(k) \tag{58}$$

where $\boldsymbol{\lambda}$ is the state transfer matrix of vehicle motion, $\hbar$ is the control matrix of vehicle motion, and $\overline{N}(k)$ is the Gaussian noise matrix generated during the vehicle motion.

$$\boldsymbol{\lambda} = \begin{bmatrix} \boldsymbol{\lambda} & & \\ & 1 & \tau \\ & & 1 \end{bmatrix} \tag{59}$$

$$\hbar = \begin{bmatrix} \widetilde{\hbar} \\ 0 \\ 0 \end{bmatrix} \tag{60}$$

Finally, the prior distribution $P(\varsigma_i(k)|\varsigma_i(k-1))$ in (47) is obtained according to the state transition model $\aleph(k)$ in (58).

### 4.3. RIS Phase Shift Matrix Design Based on Subspace Self-Organizing Iterations

The initial phase shift matrix of the RIS $\boldsymbol{\Phi}_0$ obtained in the previous section is only roughly designed based on the azimuthal angle of the vehicle's ultrasonic signals and BS, and its direct use for the reflection of the mmWave signals will produce large errors. Therefore, in this section, we propose a subspace self-organizing iterative method for adjusting the phase of the RIS reflection elements and finally designing the phase shift matrix.

When the RIS is unadjusted, there is a phase error between it and the desired phase distribution at optimal beamforming. We introduce a phase deviation matrix $G_{RIS} = diag(g)$ for the RIS, where $g$ is the phase deviation vector, which can be expressed as

$$g = \left[ e^{j\varphi_1}, e^{j\varphi_2}, \cdots e^{j\varphi_{N_{RIS}}} \right] \tag{61}$$

where $\varphi_1, \varphi_2, \cdots \varphi_{N_{RIS}}$ is the angular deviation to be adjusted for each RIS.

At this point, the signal arriving at the RIS is expressed as

$$\begin{aligned} Y_{RIS} &= G_{RIS}A_{RIS}H_{BR}F_{all}s_2 + N \\ &= G_{RIS}A_{RIS}S + N \end{aligned} \tag{62}$$

where $A_{RIS} = \left[ a_{RIS}(\bar{\theta}_1), a_{RIS}(\bar{\theta}_2), \ldots, a_{RIS}(\bar{\theta}_{N_{RIS}}) \right] \in \mathbb{C}^{N_{RIS} \times N_{RIS}}$ is the array flow pattern matrix in one-dimensional angle of RIS, $a_{RIS}(\bar{\theta}_i) = \left[ 1, e^{-j\frac{2\pi d}{\lambda}\sin\bar{\theta}_i}, \ldots, e^{-j\frac{2\pi(N_{RIS}-1)d}{\lambda}\sin\bar{\theta}_i} \right]^T$ is the array oriented vector in one-dimensional angle, and $S = H_{BR}F_{all}s_2$ is the equivalent signal vector.

The autocorrelation matrix of the signal arriving at the RIS can be expressed as

$$\begin{aligned} \mathbf{R}_{Y_{RIS}} &= \mathbb{E}[Y_{RIS}Y_{RIS}^H] \\ &= (G_{RIS}A_{RIS})E[SS^H](G_{RIS}A_{RIS})^H + E[NN^H] \\ &= (G_{RIS}A_{RIS})\mathbf{R}_S(G_{RIS}A_{RIS})^H + \mathbf{R}_N \end{aligned} \tag{63}$$

where $\mathbf{R}_S$ is the signal covariance matrix and $\mathbf{R}_N$ is the noise covariance matrix. Eigenvalue decomposition of the autocorrelation matrix can be expressed as:

$$\mathbf{R}_{Y_{RIS}} = U\Pi U^H \tag{64}$$

where $\Pi$ is the diagonal matrix consisting of eigenvalues $\lambda_1, \lambda_2, \cdots, \lambda_{N_{RIS}}$ and $U$ is the matrix consisting of eigenvectors corresponding to the eigenvalues.

The eigenvalues are ordered from largest to smallest, then the first $N_s$ corresponding eigenvectors of the eigenvalues form the signal subspace $U_s$, and the next $N_{RIS} - N_s$ eigenvectors form the noise subspace $U_n$. The diagonal matrices of the corresponding eigenvalues are $\Pi_s$ and $\Pi_n$. Therefore, (49) can be expressed as

$$\mathbf{R}_{Y_{RIS}} = U_s \Pi_s U_s^H + U_n \Pi_n U_n^H \tag{65}$$

Comparing (63) and (65), the signal subspace $U_s$ is linearly related to the RIS array prevalence matrix in the presence of phase deviation. Thus, the $U_s$ can be expressed as

$$U_s = G_{RIS} A_{RIS} Z \tag{66}$$

where $Z$ is the conversion matrix between $U_s$ and $G_{RIS} A_{RIS}$.

According to the MUSIC algorithm [40], the signal subspace and noise subspace are orthogonal. According to the above analysis, orthogonality exists between $G_{RIS} A_{RIS}$ and the noise subspace $U_s$. Therefore, we define a cost function:

$$
\begin{aligned}
\overline{\Theta} &= \left\| U_n^H G_{RIS} A_{RIS} \right\|_F^2 \\
&= \sum_{i=1}^{N_{RIS}} \left\| U_n^H G_{RIS} a_{RIS}(\overline{\theta}_i) \right\|^2 \\
&= \sum_{i=1}^{N_{RIS}} \left\| U_n^H \mathbf{O}(\overline{\theta}_i) g^H \right\|^2 \\
&= g \left( \sum_{i=1}^{N_{RIS}} U_n^H \mathbf{O}(\overline{\theta}_i) \mathbf{O}^H(\overline{\theta}_i) U_n \right) g^H
\end{aligned}
\tag{67}
$$

where $\mathbf{O}(\overline{\theta}_i) = diag\{ a_{RIS}(\overline{\theta}_i) \}$.

The idea of the proposed RIS phase shift matrix design with self-organized iteration in subspace is to minimize the cost function as a criterion. One reflection element is fixed while adjusting the other elements from the first RIS reflection element. The final RIS phase shift matrix is obtained by iterating sequentially. Thus, the problem to be solved can be formulated as follows.

$$
\begin{aligned}
\min \quad &\overline{\Theta} \\
\text{s.t.} \quad &g_n \left( e^{[n]} \right)^H = 1
\end{aligned}
\tag{68}
$$

where $e^{[n]}$ denotes the unit column vector with the $n$-th vector being 1, and $g_n$ denotes the phase deviation vector when the $n$-th reflection element is fixed. In this process, $g_n$ is calculated as follows.

$$
g_n = \frac{\left( \sum_{i=1}^{N_{RIS}} U_n^H \mathbf{O}(\overline{\theta}_i) \mathbf{O}^H(\overline{\theta}_i) U_n \right)^{-1} e^{[n]}}{\left( e^{[n]} \right)^H \left( \sum_{i=1}^{N_{RIS}} U_n^H \mathbf{O}(\overline{\theta}_i) \mathbf{O}^H(\overline{\theta}_i) U_n \right)^{-1} e^{[n]}}
\tag{69}
$$

Next, the steps of the proposed subspace self-organized iterative RIS phase shift matrix design are shown in Algorithm 1:

---

**Algorithm 1:** RIS phase shift matrix design based on subspace self-organizing iterations.

---

1. **Input:** A rough initial phase shift matrix for RIS $\boldsymbol{\Phi}_0$
2. Eigenvalue decomposition to obtain signal subspace $\boldsymbol{U}_s$ and noise subspace $\boldsymbol{U}_n$
3. $n = 0$
4. **while** $\overline{\boldsymbol{\Theta}}_n - \overline{\boldsymbol{\Theta}}_{n-1} < \varepsilon$ do
5. Use each phase value in $\boldsymbol{\Phi}_n$ as the initial value, fix the first RIS cell
6. Calculate $\boldsymbol{g}_{n+1}$ to obtain $\boldsymbol{G}_{RIS}^{(n+1)} = diag\left(\boldsymbol{g}_{n+1}\right)$ by (48)
7. Design a new phase shift matrix as $\boldsymbol{\Phi}_{n+1} = \boldsymbol{G}_{RIS}^{(n+1)} \boldsymbol{\Phi}_n$
8. Calculate cost function $\overline{\boldsymbol{\Theta}}_{n+1}$
9. $n = n + 1$
10. Repeat the step4 to step9 until convergence occurs
11. **Output:** $\boldsymbol{\Phi}_n$

---

In the proposed subspace self-organizing iterative RIS phase shift matrix algorithm, the phase shift matrix will converge to the optimal solution after multiple iterations. And the iteration ends after meeting the threshold conditions. In addition, since our algorithm starts from the initial phase of the RIS obtained with ultrasound assistance, it does not start from a random phase value. This is beneficial as it reduces the number of iterations and improves the performance of the algorithm.

## 5. BS Beamforming Matrix Design

With the phase shift matrix of RIS $\boldsymbol{\Phi}$ fixed, according to (37), the beamforming matrix of BS reaches the optimal solution when $\text{tr}\left(\boldsymbol{F}_{all}\boldsymbol{F}_{all}^H\right) = P_{\max}$ is satisfied, i.e., BS maintains the maximum transmit power. Based on the SVD, the cascaded channel $\boldsymbol{H}_{all}$ can be expressed as

$$
\begin{aligned}
\boldsymbol{H}_{all} \quad &= \boldsymbol{\Lambda}\boldsymbol{\Sigma}\boldsymbol{V}^H \\
&= \boldsymbol{\Lambda}
\begin{bmatrix}
\Sigma_1 & & & & \\
& \ddots & & & \\
& & \Sigma_{n_s} & & \\
& & & \ddots & \\
& & & & \Sigma_{N_s}
\end{bmatrix}
\boldsymbol{V}^H
\end{aligned}
\tag{70}
$$

where $\boldsymbol{\Lambda} \in \mathbb{C}^{N_r \times N_s}$ is the left singular matrix, $\boldsymbol{V} \in \mathbb{C}^{N_t \times N_s}$ is the right singular matrix, and $\Sigma_1, \cdots, \Sigma_{N_s}$ is the singular value obtained from the decomposition of $\boldsymbol{H}_{all}$. If $\boldsymbol{H}_{all}$ is given, the optimal beamforming matrix of BS can be designed as

$$
\boldsymbol{F}_{all} = \boldsymbol{\Lambda}\boldsymbol{P}^{\frac{1}{2}}
\tag{71}
$$

where $\boldsymbol{P} \triangleq diag(\boldsymbol{p}) \in \mathbb{C}^{N_s \times N_s}$ is the power allocation matrix obtained according to the water injection power allocation method [38], $p_{n_s}$ is the best allocated power for the $n_s$-th transmit data stream, and $\sum_{i=1}^{N_s} p_i = P_{\max}$.

## 6. Tensor-Based Design of Vehicle Combination Matrix

We can extend the time dimension of the RIS-vehicle channel matrix in (34) by expressing the tensor form as follows:

$$
\begin{aligned}
\boldsymbol{\mathcal{G}} &= \sqrt{\frac{N_r N_{RIS}}{L_{RU}}} \sqrt{A_{RU}} \sum_{l=1}^{L_{RU}} \boldsymbol{a}_{UE}(\phi_{RU,r}^{(l)}, \theta_{RU,r}^{(l)}) \circ \boldsymbol{a}_{RIS}^C(\phi_{RU,t}^{(l)}, \theta_{RU,t}^{(l)}) \circ \widetilde{\boldsymbol{\beta}}_l \\
&= \sqrt{\frac{N_r N_{RIS} A_{RU}}{L_{RU}}} \cdot \left(\boldsymbol{\mathcal{I}} \times_1 \boldsymbol{A}_U \times_2 \boldsymbol{A}_R^C \times_3 \boldsymbol{\Xi}_{\mathcal{G}}\right)
\end{aligned}
\tag{72}
$$

where $\widetilde{\boldsymbol{\beta}}_l = [\beta_l(1), \ldots, \beta_l(T)]^T \in \mathbb{C}^{T \times 1}$ is the component vector of the gain under each path containing the Doppler shift; $\mathcal{G}$ is the third-order channel tensor of the RIS-vehicle; and its three factor matrices are defined as

$$A_U = [\boldsymbol{a}_{UE}((\phi^{(1)}_{RU,r}, \theta^{(l)}_{RU,r}), \ldots, \boldsymbol{a}_{UE}((\phi^{(L_{RU})}_{RU,r}, \theta^{(L_{RU})}_{RU,r}))] \in \mathbb{C}^{N_r \times L_{RU}} \tag{73}$$

$$A_R = [\boldsymbol{a}_{RIS}(\phi^{(1)}_{RU,t}, \theta^{(1)}_{RU,t}), \ldots, \boldsymbol{a}_{RIS}(\phi^{(L_{RU})}_{RU,t}, \theta^{(L_{RU})}_{RU,t})] \in \mathbb{C}^{N_{RIS} \times L_{RU}} \tag{74}$$

$$\Xi_{\mathcal{G}} = [\widetilde{\boldsymbol{\beta}}_1, \ldots, \widetilde{\boldsymbol{\beta}}_{L_{RU}}] \in \mathbb{C}^{T \times L_{RU}} \tag{75}$$

Then, the received signal $Y(k)$ in (29) can be expressed as

$$Y(k) = \sqrt{\frac{P_t}{N_s}} W^H_{all} H_{RU} \widetilde{F}_{all} s_2 + N \tag{76}$$

$$\widetilde{F}_{all} = \Phi H_{BR} F_{all} = \widetilde{F}_{RF} \widetilde{F}_{BB} \tag{77}$$

Based on the above transformation, we can consider $\widetilde{F}_{all}$ as the equivalent beamforming matrix. $\widetilde{F}_{BB}$ and $\widetilde{F}_{RF}$ are the baseband equivalent digital beamformer and analogue equivalent beamformer, respectively. Then, the tensor form of the received signal $Y(k)$ can be derived from the channel tensor $\mathcal{G}$ of the RIS- vehicle, i.e.,

$$\begin{aligned} \overline{\mathcal{Y}} &= \sqrt{\frac{P_t}{N_s}} (\mathcal{G} \times_1 W^H_{all} \times_2 \widetilde{F}^T_{all} s_2) + \overline{\mathcal{N}} \\ &= \sqrt{\frac{P_t}{N_s}} (\mathcal{I} \times_1 W^H_{all} A_U \times_2 \widetilde{F}^T_{all} A^C_R s_2 \times_3 \Xi_{\mathcal{G}}) + \overline{\mathcal{N}} \end{aligned} \tag{78}$$

When the beamforming matrix of BS $F_{all}$ and the phase shift matrix of RIS $\Phi$ are determined, $\widetilde{F}_{all}$ is a known quantity. At this moment, the digital combiner $W_{BB}$ by using the MMSE criterion is represented as

$$(W_{BB})^H = \mathbb{E}[s_2 \overline{\boldsymbol{y}}^H(n)] \mathbb{E}[\overline{\boldsymbol{y}}(n) \overline{\boldsymbol{y}}^H(n)]^{-1} \tag{79}$$

where $\overline{\boldsymbol{y}}(n) \in \mathbb{C}^{M_r \times 1}$ is a column tensor of the tensor $\overline{\mathcal{Y}}$, $\mathbb{E}[s_2 \overline{\boldsymbol{y}}^H(n)]$ is the intercorrelation matrix between $s_2$ and $\overline{\boldsymbol{y}}^H(n)$, and $\mathbb{E}[\overline{\boldsymbol{y}}(n) \overline{\boldsymbol{y}}^H(n)]$ is the autocorrelation matrix of $\overline{\boldsymbol{y}}(n)$. Therefore, $\mathbb{E}[s_2 \overline{\boldsymbol{y}}^H(n)]$ and $\mathbb{E}[\overline{\boldsymbol{y}}(n) \overline{\boldsymbol{y}}^H(n)]$ can be expressed as follows, respectively,

$$\mathbb{E}[s_2 \overline{\boldsymbol{y}}^H(n)] = \frac{P_t}{N_s} \widetilde{F}^H_{BB} \widehat{H}^H_{RF}(n) \tag{80}$$

$$\mathbb{E}[\overline{\boldsymbol{y}}(n) \overline{\boldsymbol{y}}^H(n)] = \frac{P_t}{N_s} \widehat{H}_{RF}(n) \widetilde{F}_{BB} \widetilde{F}^H_{BB} \widehat{H}^H_{RF}(n) + \sigma^2 W^H_{RF} W_{RF} \tag{81}$$

where $\widehat{H}_{RF}(n) = W^H_{RF} H_{RU}(n) \widetilde{F}_{RF}$ is the baseband equivalent channel matrix.

In order to calculate $W_{BB}$, it is necessary to firstly obtain $\widetilde{F}_{BB}$ and $W_{RF}$. According to the multi-vehicle joint optimization method [14], the equivalent analogue beamformer $\widetilde{F}_{RF}$ and the analogue combiner for the user vehicle $W_{RF}$ can be designed. Additionally, the equivalent digital beamformer $\widetilde{F}_{BB}$ can be calculated from $\widetilde{F}_{BB} = \widetilde{F}_{all} F^{-1}_{RF}$.

Put (80) and (81) into (79); the digital combiner $W_{BB}$ can be expressed as

$$W_{BB} = \left( \widehat{H}_{RF}(n) \widetilde{F}_{BB} \widetilde{F}^H_{BB} \widehat{H}^H_{RF}(n) + \frac{N_s \sigma^2}{P_t} W^H_{RF} W_{RF} \right)^{-1} \widehat{H}_{RF}(n) \widetilde{F}_{BB} \tag{82}$$

Finally, the combination matrix of vehicle can be expressed as

$$W = W_{RF}(\widehat{H}_{RF}(n)\widetilde{F}_{BB}\widetilde{F}_{BB}^{H}\widehat{H}_{RF}^{H}(n) + \frac{N_s\sigma^2}{P_t}W_{RF}^{H}W_{RF})^{-1}\widehat{H}_{RF}(n)\widetilde{F}_{BB} \tag{83}$$

## 7. Simulation Results and Analysis

In this section, we provide simulation results to describe the proposed beamforming algorithm.

### 7.1. Simulation Results of FSO Link

During the simulation, we evaluate the BER and outage probability of the FSO link. For the FSO link, the channel fading is considered to be a Gamma–Gamma stochastic process. In this section, the effects of pointing errors and different turbulence strengths on the performance of the FSO link are investigated for different values of $\varsigma$. According to [41], we set the parameters as follows: the laser wavelength is $\lambda_{FSO}$ = 1550 nm, the beam divergence angle is $\theta$ = 2 mrad, the receiver diameter is 20 cm and the receiver aperture radius is 6 cm.

Figure 6 depicts the BER of the FSO link for different strengths of atmospheric turbulence with $\alpha = 2.289$, $\beta = 1.834$ (moderate turbulence) and $\alpha = 2.052$, $\beta = 1.335$ (strong turbulence). Based on the previous derivation in the FSO system model, it is clear that $\varsigma$ is inversely proportional to the pointing error displacement standard deviation $\sigma_s$ and the pointing error is larger when $\varsigma$= 1.5. In Figure 6, an increase in the pointing error leads to a significant increase in the BER of FSO link for the same turbulent conditions. For the same pointing errors, strong turbulence has a greater impact on the BER performance of the FSO link. This is because strong turbulence causes large attenuation and phase disturbance during signals transmission, which affects the BER of the FSO link. It is worth noting that larger pointing errors lead to lower BER performance of the FSO link. However, in the case of slight misalignment, this can be considered negligible.

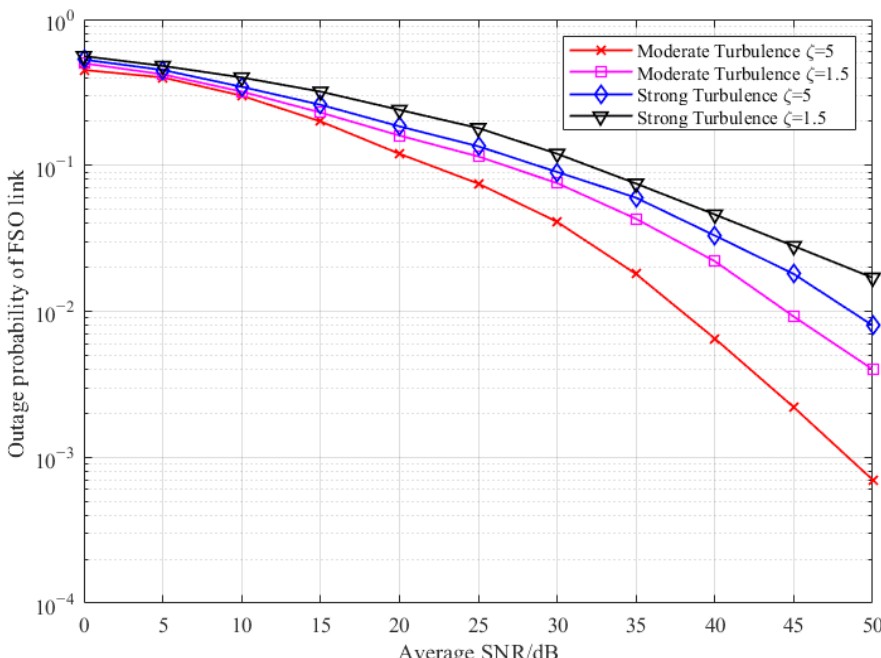

**Figure 6.** BER performance of FSO link with different turbulence strengths and pointing errors.

Figure 7 depicts the effect of pointing error on the FSO link outage probability for several of the same atmospheric turbulence conditions as those described in Figure 6. We note that the outage probability increases with the strength of the atmospheric turbulence

and the severity of the pointing error. In fact, for the same turbulence strengths, the pointing error significantly affects the performance of the FSO link. In the moderate turbulence case, from $\varsigma = 5$ to $\varsigma = 1.5$, the outage probability of FSO link increases significantly; the same result is found for strong turbulence, where the outage probability is proportional to the severity of the pointing error. And the effect is somewhat larger in the case of strong turbulence.

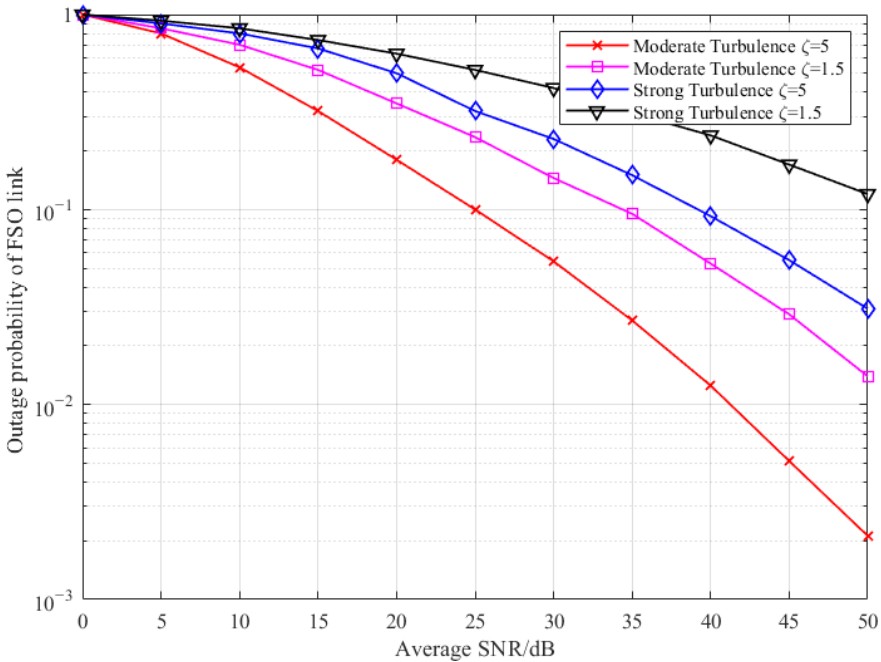

**Figure 7.** Outage probability of FSO link with different turbulence strengths and pointing errors.

### 7.2. Simulation Results of V2X

In the simulation, according to [29,30], we set the mmWave massive MIMO communication system to operate in the 73 GHz band, the number of transmit antennas at the BS is set to $N_t = 16$, the number of RF chains is $M_t = 16$, the number of receive antennas at the vehicle is $N_r = 4$, and the number of receive RF chains is $M_r = 4$. The antenna spacing is a half-wavelength length, the number of RIS elements is $N_{RIS} = 64$, the number of complex gain propagation paths is $L_{BR} = L_{RU} = 4$, and the number of data streams is $N_s = 4$. The downlink uses the following transmission frame structure: $T_1 = 120$, $T_2 = 5$, $T_3 = 45$. The azimuths of AOA and AOD are uniformly distributed in the interval $[-\pi, \pi)$ and the elevation angles of AOA and AOD are uniformly distributed in the interval $\left[-\frac{\pi}{2}, \frac{\pi}{2}\right)$. The maximum Doppler shift is $f_{\max}$ and the noise power is $\sigma^2 = -80$ dBm. The speed of the vehicle is set to 0–150 km/h and the range of exponential factors $\omega$ is [1/1024, 7/512, 9/256, 15/128, 30/64, 50/32].

In this section, we use the spectral efficiency described in [42] and the defined signal-to-noise ratio (SNR) in [32] as evaluation metrics; the SNR is expressed as

$$SNR = \frac{P_t \left\| \mathbf{W}_{all}^H \mathbf{H}_{all} \mathbf{F}_{all} \right\|_F^2}{\sigma^2 N_s} \tag{84}$$

Considering the ideal phase shift, the proposed method is compared with the No RIS, random phase shift, orthogonal matched tracking (OMT) and sum path gain maximization–semipositive definite relaxation (SPGM-SDR) to obtain the following simulation results. At the same time, the DOA tracking algorithm proposed in this paper is compared with conventional likelihood PF (CL-PF), projection approximation subspace tracking (PAST),

modified PAST (MPAST), and PAST with deflation (PASTD). The root mean square error (RMSE) is used to evaluate the tracking accuracy of the DOA tracking algorithm.

$$RMSE = \frac{1}{N_R} \sum_{m=1}^{N_R} \sqrt{[\psi_m(n) - \overline{\psi}_m(n)]^2} \tag{85}$$

where $N_R$ is the total number of ultrasonic array elements, $\psi_m(n)$ is the estimated value of DOA tracking in the $n$-th time slot, and $\overline{\psi}_m(n)$ is the actual value of DOA tracking in the $n$-th time slot.

DOA tracking is important in communication systems, as it is used to adjust beamforming and improve system performance. We compared the proposed ultrasonic-assisted DOA tracking algorithm with others. In Figure 8, it can be seen that PAST, MPAST and PASTD have larger errors at some estimation points because these algorithms are based on updated subspace. It is more difficult to guarantee the orthogonality of the signal and noise subspace during the update process. The performance of PF-MUSIC is close to that of our proposed algorithm, but PF-MUSIC requires a large number of eigenvalue decompositions and suffers from high complexity and high computational power. Our algorithm maintains good RMSE performance at the same SNR. In summary, the ultrasonic-assisted DOA tracking algorithm proposed in this paper has better results than other algorithms, which can provide more accurate phases for the phase shift matrix of RIS.

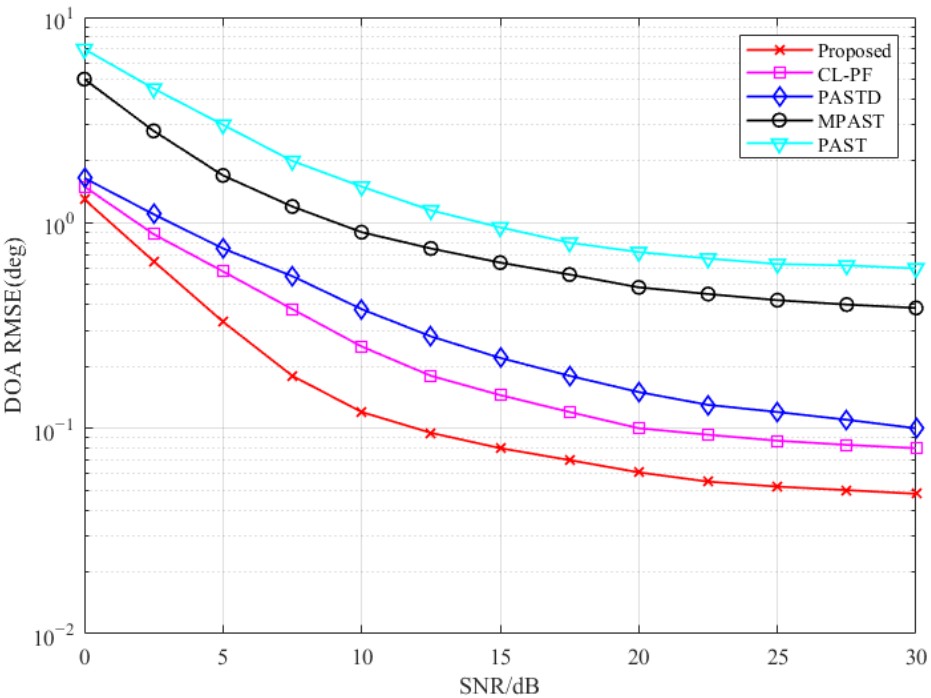

**Figure 8.** The performance of DOA tracking with different algorithms.

Figure 9 depicts the relationship between spectral efficiency and transmit power of BS. The overall spectral efficiency of No RIS is much lower than the others, indicating that the signal transmission of the No RIS system is more affected by the transmission route. The other algorithms improve the capacity of the system more obviously. The random phase shift is less effective because the accuracy of the phase shift angle cannot be guaranteed. The performance of SPGM-SDR and OMP is similar. The proposed algorithm has the best spectral efficiency, and the results are slightly higher those of than SPGM-SDR and OMP. This proves the effectiveness of the proposed tensor-based ultrasonic and RIS-assisted beamforming, and the RIS also can improve the spectral efficiency of the system.

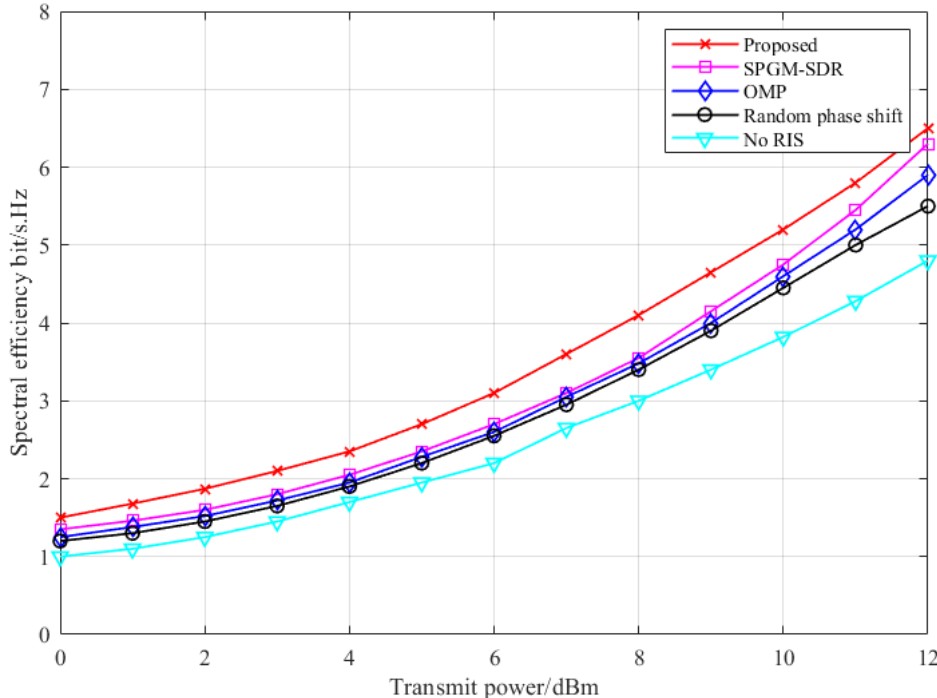

**Figure 9.** The relationship between spectral efficiency and transmission power.

Figure 10 shows the spectral efficiency versus SNR. From Figure 10, it can be seen that no RIS has the worst performance because the direct path between the BS and vehicle is obscured, resulting in a large loss of signal reaching the vehicle. The random phase shift can establish communication with the vehicle through RIS more effectively, which improves the spectral efficiency compared to that of No RIS. But the phase shift angle cannot be updated effectively and performs less well than other three algorithms. The proposed algorithm has higher spectral efficiency than the SPGM-SDR and OMP because the proposed algorithm considers the effect of vehicle's mobility, and the ultrasonic-assisted RIS phase shift angle self-organized is updated, which is more helpful for the update of the RIS phase shift matrix and has better spectral efficiency.

The performance of the RIS-assisted communication system greatly depends on the number of reflecting elements. To investigate the dependency between them, we simulate the spectral efficiency performance of the proposed algorithm for different numbers of RIS reflecting elements. Figure 11 shows the trend of spectrum efficiency with the number of RISs' reflecting elements for five algorithms. As the number of RISs' reflective elements increases, the spectral efficiency of all the algorithms except No RIS is significantly improved. The proposed algorithm always achieves the highest spectral efficiency compared with the chosen baseline algorithms. SPGM-SDR can only approximate the spectral efficiency without reaching the optimal result. While the proposed algorithm can obtain a better solution via a joint iterative update, the proposed algorithm can obtain significant improvement in spectral efficiency.

In Figure 12, we investigate the number of RISs' reflecting elements on the BER. As the number of RISs' reflecting elements increases, the BER can be improved at the same speed. RIS can improve signal strength and quality by focusing the signal on the target receiver through beamforming and can also reduce signal impairment. This indicates that RIS- assisted V2X communication can improve the system's performance.

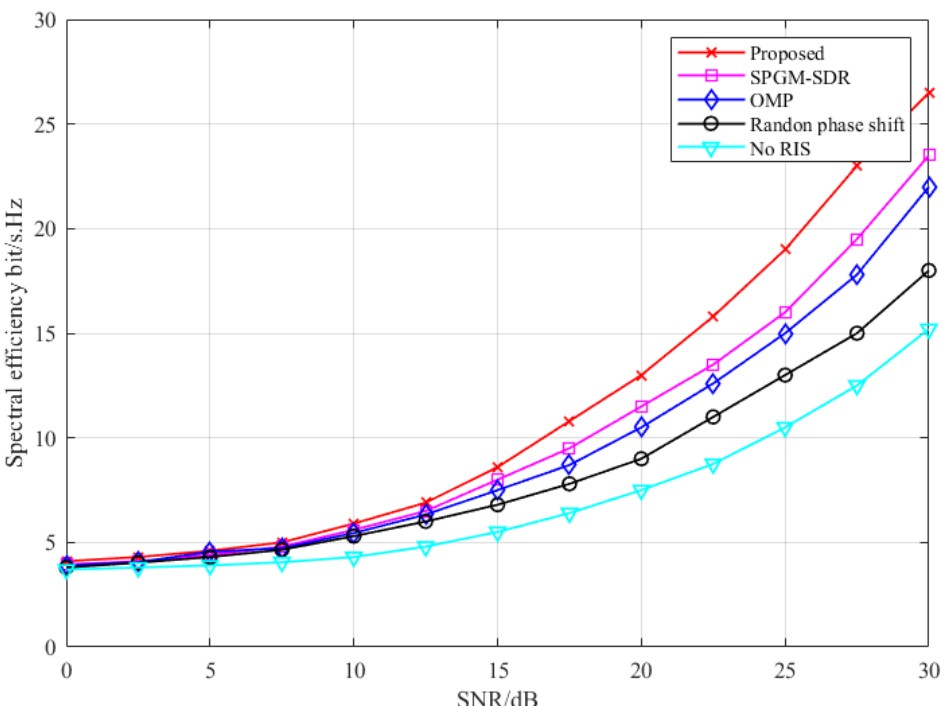

**Figure 10.** The relationship between spectral efficiency and SNR.

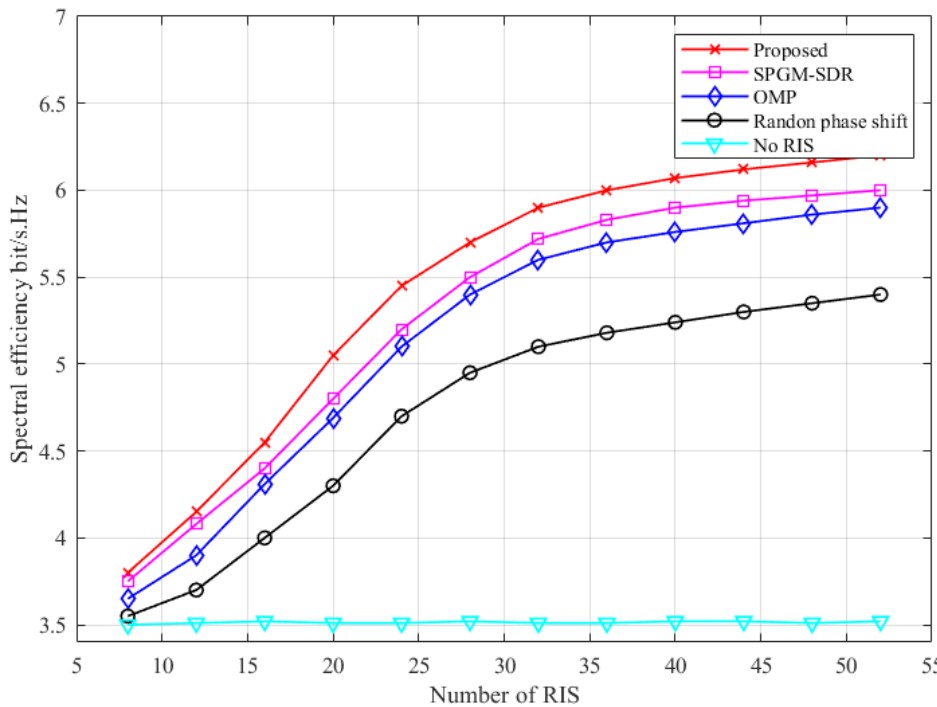

**Figure 11.** The effect of the number of RISs' reflecting elements on the spectral efficiency.

In V2X communication, Doppler shift will affect the communication performance of the system. Figure 13 shows the BER versus the maximum Doppler shift caused by the change of vehicle speed. It can be seen that the BER of the communication system is related to the Doppler shift and deteriorates with the increase in Doppler shift. A larger Doppler shift at higher vehicle speeds leads to frequency deviation of the received echo signal, which affects the effectiveness of beamforming. Compared with other algorithms, the proposed algorithm still maintains good stability and robustness because the proposed algorithm considers the time-varying channel with Doppler shift and uses ultrasonic-assisted DOA

tracking and beam angle update. This shows that the proposed beamforming algorithm has better performance.

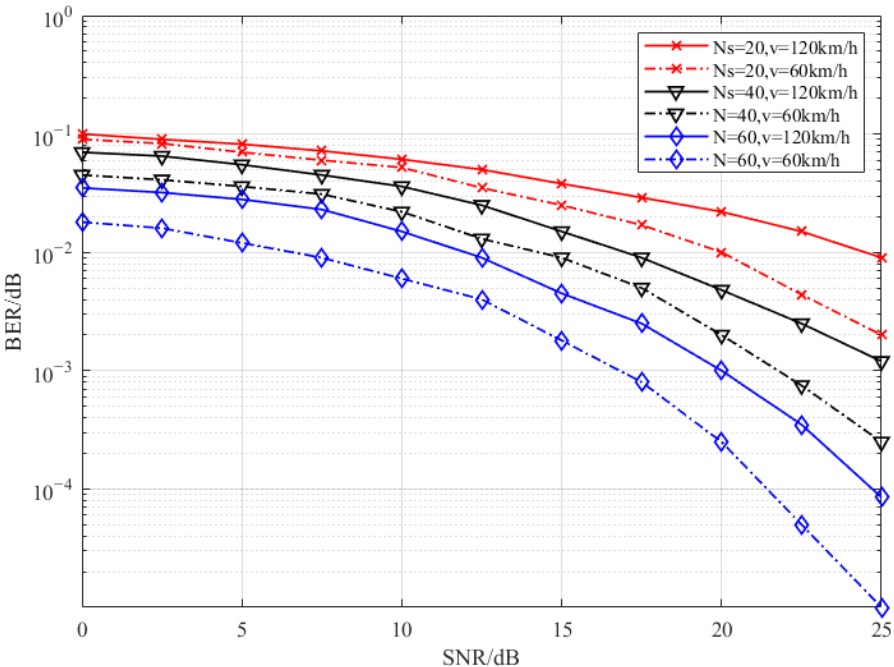

**Figure 12.** The performance of different vehicle speeds and the number of RISs' reflecting elements.

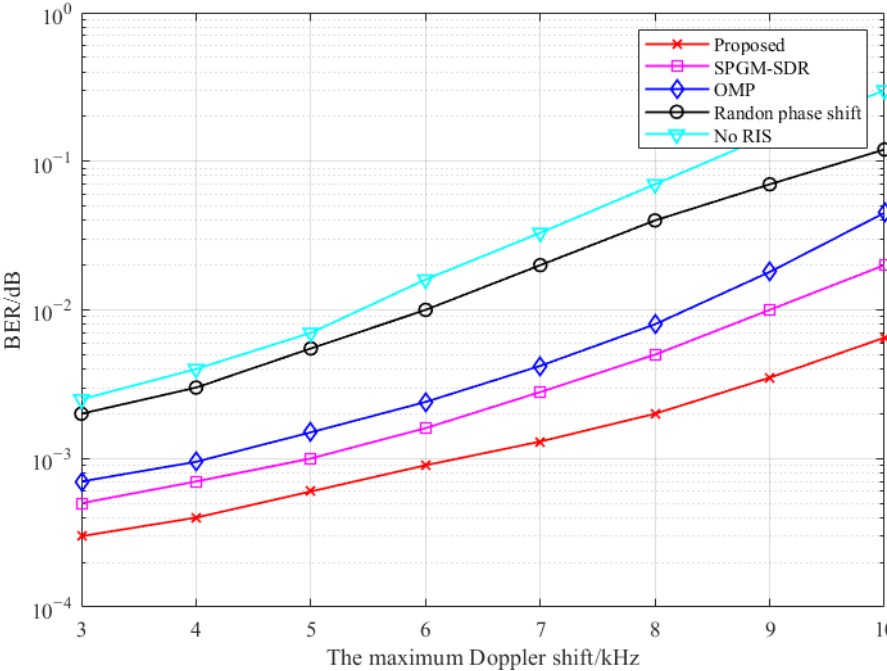

**Figure 13.** The effect of the maximum Doppler shift on the performance of communication systems.

## 8. Discussion

The results of this paper show that the proposed algorithm can improve the spectral efficiency of the communication system to a certain extent. It has lower BER compared to other algorithms. The goal of maximizing the analysis via spectral efficiency is achieved. The communication quality can be improved by increasing the number of RISs' reflecting elements, and the proposed algorithm has better performance in V2X communication because it takes into account the existence of the Doppler shift problem in V2X communication. The

proposed algorithm effectively solves the problems of occlusion in LoS communication, the accurate DOA estimation to V2X communication and the difficulty in solving the rank corresponding to CP decomposition. The V2X communication performance is improved.

V2X communication is currently very popular, and RIS-assisted V2X research is in its infancy. Compared with similar studies [28–30], our study discussed the communication scenario of massive MIMO V2X, but our study also considered the use of RIS-assisted communication, which is able to solve the problem of occlusion in the LoS path and improve the performance to some extent. Deep learning and artificial intelligence are now very popular in the communications field, and combining them with RIS-assisted V2X is also very novel. Therefore, our future research will be directed towards this area. The RIS phase shift matrix design problem can be better solved via deep learning.

## 9. Conclusions

In this study, we aimed to address the problems with mmWave massive MIMO V2X due to obstacle occlusion and mmWave transmission range limitation. Combining the advantages of FSO and RIS, a tensor-based joint beamforming scheme for ultrasonic and RIS-assisted for dual-hop hybrid FSO-mmWave massive MIMO of V2X was proposed. The V2X communication range was increased by the FSO link, and the problem of occlusion in vehicle LoS communication was solved using RIS-assisted V2X. Then, we utilized ultrasonic and RIS-assisted mmWave massive MIMO V2X. The ultrasonic-assisted RIS phase shift matrix was designed based on subspace self-organizing iterations based on the DOA information obtained from ultrasonics. The beam direction between the RIS and the vehicle during vehicle motion was tracked. Finally, the Tucker tensor decomposition was used to describe the high-dimensional beam space, and a cascaded channel decomposition method based on SVD was adopted to obtain the combined matrix beamforming combination matrix for BS and vehicle, which solved the problem of difficult rank solution corresponding to the CP tensor decomposition. The simulation results show that the spectral efficiency of the proposed method significantly improved, and the BER was lower compared with other methods. And it can solve the above problem very well.

**Author Contributions:** Conceptualization, X.Z. and Z.Z.; methodology, X.Z.; software, Z.Z. and J.L.; validation, X.Z. and L.T.; formal analysis, X.Z. and Z.M.; investigation, X.Z. and L.T.; resources, X.Z. and J.L.; data curation, J.L.; writing—original draft preparation, Z.Z.; writing—review and editing, Z.Z.; visualization, J.L.; supervision, X.Z.; project administration, X.Z. and Z.M. All authors have read and agreed to the published version of the manuscript.

**Funding:** This research was funded by Shanghai Capacity Building Projects in Local Institutions, grant number 19070502900, Science and Technology Commission of Shanghai Municipality, grant number 22142201900.

**Institutional Review Board Statement:** Not applicable.

**Informed Consent Statement:** Not applicable.

**Data Availability Statement:** Not applicable.

**Conflicts of Interest:** The authors declare no conflict of interest.

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
