# Peer review of "Tensor-Based Joint Beamforming with Ultrasonic and RIS-Assisted Dual-Hop Hybrid FSO mmWave Massive MIMO of V2X"

_photonics, doi:10.3390/photonics10080880_

Round 1

Reviewer 1 Report

This paper can be accepted. The work is interesting and adequately addresses the problem. The combination of FSO and MIMO is novel and can improve the communication range. Ultrasonic can largely improve the accuracy of DOA estimation. RIS can improve the quality of mmWave massive MIMO IoV communications, and tensor can efficiently handle high-dimensional signals. The work is a good solution to the problem of IoV communication being blocked. However, in order to improve further the quality of the manuscript, the author may consider to incorporate following comments:

1.      The destination node (D) in the system model of Figure 1 should be explained before the section 2.1.

2.      in Equation (4) is inconsistent with the previously mentioned the optical signal vector .

3.       in Equation (55) is not defined.

4.       in Equation (56) is not defined.

Author Response

Dear Editor/Reviewers:

Thank you very much for your extraordinary efforts in reviewing our paper! The comments received are precious, from which we can feel that all reviewers well appreciate the theoretical value of this work for allowing a resubmission of our manuscript, with an opportunity to address the reviewers’ comments.

We have carefully revised it by the reviewers' comments as well as corrected some details identified during the revision process.

Point 1: The destination node (D) in the system model of Figure 1 should be explained before the section 2.1.

Response 1: Thanks for your comments. We have added an explanation of the destination node (D) before the section 2.1.

Point 2: $s_1$ in Equation (4) is inconsistent with the previously mentioned the optical signal vector .

Response 2: Thanks for your comments. We have unified into $s_1$.

Point 3: $\overset{\scriptscriptstyle\frown}{H}$ in Equation (55) is not defined.

Response3: Thanks for your comments. The $\overset{\scriptscriptstyle\frown}{H}$  should be replaced by $H_{RU}$.

Point 4: $\overset{\scriptscriptstyle\smile}{H}$ in Equation (56) is not defined.

Response 4: Thanks for your comments. The $\overset{\scriptscriptstyle\smile}{H}$ should be replaced by $H_{BR}$.

Reviewer 2 Report

This paper is ready to be accepted. The author have well explained the proposed topic. This paper presented a beamforming method to address blocked IoV communications, which uses ultrasonic-assisted vehicle DOA estimation to provide accurate RIS phase for RIS-assisted IoV communications. And tensor has certain advantages for processing high-dimensional signals. It also combined the advantages of FSO to improve communication range and safety. However, some of these issues need to be addressed as follows:

1.       Figure 2 should be positioned before the section 2.1, since it cannot be seen and read properly.

2.       The power fraction  in section 2.1 is inconsistent with  in equation (1).

3.       The number of narrowband ultrasonic signals  in section 2.4 is inconsistent with  in .

4.       The form of the annotation in Figure 3 should be the same as the following figures.

Minor English editing is needed . 

Author Response

Dear Editor/Reviewers:

Thank you very much for your extraordinary efforts in reviewing our paper! The comments received are precious, from which we can feel that all reviewers well appreciate the theoretical value of this work for allowing a resubmission of our manuscript, with an opportunity to address the reviewers’ comments.

We have carefully revised it by the reviewers' comments as well as corrected some details identified during the revision process.

Point 1: Figure 2 should be positioned before the section 2.1, since it cannot be seen and read properly.

Response 1: Thanks for your comments. We have placed Figure 2 before the section 2.1

Point 2: The power fraction $M_0$ in section 2.1 is inconsistent with $M_o$ in equation (1).

Response 2: Thanks for your comments. We have unified these two symbols into $M_o$.

Point 3: The number of narrowband ultrasonic signals $N_v$ in section 2.4 is inconsistent with  $N_V$ in $\Psi $ .

Response 3: Thanks for your comments. We have unified these two symbols into $N_V $.

Point 4: The form of the annotation in Figure 3 should be the same as the following figures.

Response 4:  Thanks for your comments. We have modified the annotation of Figure 3.

Reviewer 3 Report

    This manuscript, entitled “Tensor-based joint beamforming with ultrasonic and RIS-assisted for Dual-Hop Hybrid FSO-mmWave massive MIMO of IoV,” considers an FSO-mmWave massive MIMO system assisted by ultrasonic and RIS algorithm for infrastructure-vehicle communications. The manuscript is dedicated to deriving a novel vehicle tracking algorithm, of which numerical calculations show that the proposed scheme outperforms the existing schemes. However, no numerical calculation deals with the FSO links, although the manuscript's subject is FSO-mmWave massive MIMO system. The performance of FSO links would affect the entire performance. So, the authors should have considered the dependencies of link performance on the parameters of FSO links, such as the refractive constant parameter. Therefore, I cannot recommend publishing the manuscript as a regular paper in the journal. The authors should resubmit the manuscript after adding the numerical simulation on the FSO-link performances.

    I listed below the comments on the manuscript. I hope the authors modify the manuscript according to these comments when resubmitting.

1.         Introduction contains a lot of related works. However, it is too long to grasp the novelty and impact of the manuscript. I recommend the authors make Introduction more concise.

2.         In Line 235, the statement “Where $P_t$ is the transmit.” does not make sense.

3. The main body has two similar abbreviations, “AOD” and “DOA”. They should be unified.

4.         In Lines 291-293, the authors state, “In order to simplify the above problem, we can solve the problem by alternating optimization, i.e., fixing several other matrices when solving one matrix, followed by alternating updates of the matrices to be solved in several iterations.” Does this mean that the proposed algorithm is just suboptimal? If so, what is the gap between the optimal algorithm and the proposed one?

5.         Section 4 is too long to understand the novelty. The author should stress what they have newly revealed in the manuscript. Especially in subsection 4.1, I cannot understand why the matrix for 2-D movement is simpler than that for 1-D.

6.         Eq. (25) has a parameter $\omega$ that is determined from the experimental data. How was it in the numerical simulations?

7.         In Line 369, what are matrix A and the angular information B? Besides them, the manuscript includes several undefined signs. For example, I cannot find the definition of $\beta_q$ in Eq. (38).

8.         In the numerical simulation, the authors set the vehicle as 120 km and 150 km. What is the difference between them? Plus, why are they? It seems too high for driving in the city.

I can find some typos in the manuscript. Moreover, some grammatical errors can be found. The authors should proofread the manuscript.

Author Response

Dear Editor/Reviewers:

Thank you very much for your extraordinary efforts in reviewing our paper! The comments received are precious, from which we can feel that all reviewers well appreciate the theoretical value of this work for allowing a resubmission of our manuscript, with an opportunity to address the reviewers’ comments.

We have carefully revised it by the reviewers' comments as well as corrected some details identified during the revision process. Finally we have added a discussion section.

Point 1:  Introduction contains a lot of related works. However, it is too long to grasp the novelty and impact of the manuscript. I recommend the authors make Introduction more concise.

Response 1: Thanks for your comments. We have revised the introduction as required and reduced some unnecessary content. The content of each paragraph has been changed. We changed some studies on the combination of FSO and RF in lines 31-47. In lines 48-66 we have simplified the original content about V2X and studied some system models about RIS-assisted V2X. In lines 67-83 we discussed the DOA estimation used for the RIS-assisted communication systems, and the original DOA estimation were revisited. In lines 84-93 we studied the tensor-based V2X communication system. In lines 94-116 we proposed the corresponding model according to the problems to be solved, and then we analyzed and summarized the proposed model. This part was also our main research content. Lines 117-123 provided an introduction to the content of each section. These were all the changes we have made to the introduction.

Point 2: In Line 235, the statement “Where $P_t$ is the transmit.” does not make sense.

Response 2: Thanks for your comments. The original content in Line 235 is “Where $P_t$ is the transmit.” Now we changed it to “Where $P_t$ is the transmit power.” in line 208.

Point 3: The main body has two similar abbreviations, “AOD” and “DOA”. They should be unified.

Response 3: Thanks for your comments. The meaning of these two similar abbreviations should be different, AOD means Angle of Departure, while DOA means Direction of Arrival. We have added the corresponding comments to the original text in line 65 and line 247.

Point 4: In Lines 291-293, the authors state, “In order to simplify the above problem, we can solve the problem by alternating optimization, i.e., fixing several other matrices when solving one matrix, followed by alternating updates of the matrices to be solved in several iterations.” Does this mean that the proposed algorithm is just suboptimal? If so, what is the gap between the optimal algorithm and the proposed one?

Response 4: Thanks for your comments. For the RIS-assisted MIMO V2X system, a non-convex optimization function is constructed with the objective of maximum spectral efficiency, and there is no optimal algorithm. Therefore, this paper then proposes a beamforming design based on alternating iterations using the characteristic that the qualifying conditions of the optimization function are not coupled with each other. Specifically, the scheme is based on alternating iterations to achieve suboptimal design of the RIS phase shift matrix, BS beamforming matrix and vehicle combination matrix in order to solve the optimization problem. In the design of the RIS phase shift matrix, this paper combines the subspace self-organizing iterative algorithm to transform the nonconvex optimization problem into a series of easily solvable subproblems. Numerical simulation results show that the scheme in this paper achieves a better compromise in terms of spectral efficiency improvement compared with existing schemes. We have added the relevant explanations in lines 276-287.

Point 5: Section 4 is too long to understand the novelty. The author should stress what they have newly revealed in the manuscript. Especially in subsection 4.1, I cannot understand why the matrix for 2-D movement is simpler than that for 1-D.

Response 5: Thanks for your comments. The main work in section 4 is to estimate the ultrasonic DOA arriving at the ultrasonic array and design a rough RIS initial phase shift matrix accordingly. Then the optimal RIS phase shift matrix is obtained by subspace self-organizing iterations. To better understand the content of section 4, we switched the original subsections 4.1 and 4.2. The prior distribution in subsection 4.1 requires the observation model of vehicle motion in subsection 4.2. So the subsection 4.2 is designed to solve for the observation model of vehicle motion. We illustrated this in lines 352-355. The converted 2-D motion state is used in the calculation of the prior distribution in Eq. (34). The 2-D movement is more complicated than that for 1-D, but we consider the 2-D movement in order to make the vehicle motion more realistic.

Point 6: Eq. (25) has a parameter $\omega$ that is determined from the experimental data. How was it in the numerical simulations?

Response 6: Thanks for your comments. $\omega$ is related to the number of snapshots $N_u$, according to [25], the range of exponential factors $\omega$ is [1/1024, 7/512, 9/256, 15/128, 30/64, 50/32]. We added a range of exponential factor in line 490 of the simulation data setup section.

Point 7: In Line 369, what are matrix A and the angular information B? Besides them, the manuscript includes several undefined signs. For example, I cannot find the definition of $\beta_q$ in Eq. (38).

Response 7: Thanks for your comments. Line 369 is a mistake in our writing and we have revised it. ‘’It can be seen that the matrix $\Gamma$ contains the observed information, while the angular information ${\vartheta _m}(k)$ contains a priori information’’ was the modified in line 324. The $\beta_q$ has been defined in Eq. (10). $\beta_q$ was defined as the azimuth angles.

Point 8: In the numerical simulation, the authors set the vehicle as 120 km and 150 km. What is the difference between them? Plus, why are they? It seems too high for driving in the city.

Response 8: Thanks for your comments. This paper considered the case of RIS-assisted V2X communication on a highway in lines 125-137, where the speed of the vehicle should be in the range of 0-150km/h instead of a fixed value. We have modified it in the simulation section and added a speed comparison simulation.

Reviewer 4 Report

The authors investigate the tensor-based joint beamforming with ultrasonic and RIS-assisted for Dual-Hop Hybrid FSO-mmWave massive MIMO of IoV. They constructed the dual-hop relay system considered the RIS schemes. The paper is easy to follow. However, inefficient writing skills and presentation of the results give an unacceptable interpretation. Maybe a more detailed explanation of the system model and the results obtained can make the paper effectual. The reviewer has the following concerns.

1. A typical realistic application or use case of the proposed system model is important, but it is missing in the paper.

2. The author provides a noticeable number of literature to find research gaps and motivation, which is not convincing. Highlight the research gap and main contribution of this article which is not addressed before.

3. The abstract should also be revised. The author should highlight what results have been achieved, not simply list what has been done. In addition, the abstract should be condensed for readability.

4. The literature review on the subject at hand is not complete. In addition, the literature of this work is weak, and it should be updated. There are many related works about the relay system with different channel models. The following reference (doi: 10.1109/TVT.2023.3252822, 10.1109/TAES.2022.3142116, 10.1109/JSAC.2009.091218) could help the readers and prevent misleading of previously undertaken research works.

5. The mathematical analysis in the theoretical section should combine with the physical meaning of the system model. Otherwise, the mathematical analysis is generally in any similar scenario.

6. The authors considered the Gamma-Gamma model for atmospheric turbulence, but there are also more general models (e.g., Malaga, exponential Weibull etc.)

7. The authors should justify their choice of parameters using suitable references. Much more discussion about the results should be given in this paper.

8. The presentation of all simulation results could have been better. Moreover, discussions regarding the results (Figs. 3-8) are very less than expected. Rewrite these for better understanding.

9. The article is understandable, but the grammar and spelling errors should be carefully considered in the revision.

Minor editing of English language required.

Author Response

Dear Editor/Reviewers:

Thank you very much for your extraordinary efforts in reviewing our paper! The comments received are precious, from which we can feel that all reviewers well appreciate the theoretical value of this work for allowing a resubmission of our manuscript, with an opportunity to address the reviewers’ comments.

We have carefully revised it by the reviewers' comments as well as corrected some details identified during the revision process. Finally, we have added a discussion section.

Point 1: A typical realistic application or use case of the proposed system model is important, but it is missing in the paper.

Response 1: Thanks for your comments. We have added a description of the proposed system use case in lines 125-137. An ultrasonic and RIS-assisted for hybrid FSO-mmWave massive MIMO of V2X system on a highway is proposed. We describe the use case in detail.

Point 2: The author provides a noticeable number of literatures to find research gaps and motivation, which is not convincing. Highlight the research gap and main contribution of this article which is not addressed before.

Response 2: Thanks for your comments. We have revised the introduction as required and reduced some unnecessary content. The content of each paragraph has been changed. We changed some studies on the combination of FSO and RF in lines 31-47. In lines 48-66 we have simplified the original content about V2X and studied some system models about RIS-assisted V2X. In lines 67-83 we discussed the DOA estimation used for the RIS-assisted communication systems, and the original DOA estimation are revisited. In lines 84-93 we studied the tensor-based V2X communication system. In lines 94-116 we summarized the contributions of this paper, and we proposed the corresponding model according to the problems to be solved.

Point 3: The abstract should also be revised. The author should highlight what results have been achieved, not simply list what has been done. In addition, the abstract should be condensed for readability.

Response 3: Thanks for your comments. We have revised the abstract, some of the unnecessary content we have removed. We briefly described the background of the study in 9-11. Then, we presented the problem and approach to be addressed in lines 12 to 27, and we briefly describe the main studies. The latter part contains our results.

Point 4: The literature review on the subject at hand is not complete. In addition, the literature of this work is weak, and it should be updated. There are many related works about the relay system with different channel models. The following reference (doi: 10.1109/TVT.2023.3252822, 10.1109/TAES.2022.3142116, 10.1109/JSAC.2009.091218) could help the readers and prevent misleading of previously undertaken research works.

Response 4: Thanks for your comments. We have updated some of the literature and have taken into account the references that you have mentioned. In this paper, the literature [1] described the effect of weather on the FSO/RF link, A novel downlink satellite communication (SatCom) model was proposed in literature [4], an unmanned aerial vehicle (UAV)-assisted dual-hop FSO/FSO system was proposed in literature [5].

Point 5: The mathematical analysis in the theoretical section should combine with the physical meaning of the system model. Otherwise, the mathematical analysis is generally in any similar scenario.

Response 5: Thanks for your comments. We proposed an ultrasonic and RIS-assisted for hybrid FSO-mmWave massive MIMO of V2X system on a highway in lines 125-137. Our mathematical analysis is integrated with the system model, but there are some scenarios where the mathematical analysis is similar.

Point 6: The authors considered the Gamma-Gamma model for atmospheric turbulence, but there are also more general models (e.g., Malaga, exponential Weibull etc.)

Response 6: Thanks for your comments. Now there are indeed many other models for atmospheric turbulence, we consider the Gamma-Gamma model because most of the literatures use the Gamma-Gamma model, and the atmospheric turbulence is generally a medium to strong turbulence case, the Gamma-Gamma model is more consistent with the actual test data in the turbulence case, so Gamma-Gamma model is more suitable. We added a note about the Gamma-Gamma model in lines 154-156.

Point 7: The authors should justify their choice of parameters using suitable references. Much more discussion about the results should be given in this paper.

Response 7: Thanks for your comments. In line 479 we have added references to the parameter selection in the simulation section. Then we also re-discussed the simulation results and modified some of the simulations. See the answer to point 8 for more details.

Point 8: The presentation of all simulation results could have been better. Moreover, discussions regarding the results (Figs. 3-8) are very less than expected. Rewrite these for better understanding.

Response 8: Thanks for your comments. We have optimized the simulation results and rewritten the discussion of the results. Figure 3 shows the DOA tracking performance of the proposed algorithm. Figure 4 shows the spectral efficiency versus transmit power. Figure 5 shows the spectral efficiency versus SNR. Figure 6 shows the effect of RIS reflection elements on the spectral efficiency. Figure 7 shows the effect of vehicle speed and the number of RIS reflective elements on the system BER. Figure 8 shows the effect of Doppler shift on system BER.

Point 9: The article is understandable, but the grammar and spelling errors should be carefully considered in the revision.

Response 9: Thanks for your comments. We apologized for this, and we have checked the full text for the grammar and spelling errors.

Round 2

Reviewer 3 Report

I appreciate the Authors' effort to improve the manuscript. However, they did not address my main concern, which I decided rejection. The manuscript does not contain the consideration or numerical simulation on the FSO link. They should address this point or make some explanations, or the paper should be rejected.

I have no comments on English.

Author Response

Dear Reviewer:

Thanks for your comments. We apologized for not adding a discussion of FSO links due to an oversight on our part. In the revised version, we have added a discussion of the BER and outage probability of the FSO link and derived the formulas for BER and outage probability in lines 173-202. In lines 221-262 we added a method for estimating the neural network for the FSO channel coefficient. Then, we have added numerical simulations of the FSO link BER and outage probability in lines 546 -576 and discussed the effect of atmospheric turbulence strength and pointing error on the FSO link. In the new revision, the changes have been highlighted.  Hopefully, our efforts to revise the manuscript have addressed your concerns.

Thank you again for your valuable feedback and hope to be able to get your recognition.

Reviewer 4 Report

This manuscript can be accepted this time since all my concern has been well solved.

Author Response

Thank you for your careful review and agreeing to accept our paper.